# Multicolor fate mapping of microglia reveals polyclonal proliferation, heterogeneity, and cell-cell interactions after ischemic stroke in mice

Majed Kikhia [1,2,3], Simone Schilling[1,2,4,5], Marie-Louise Herzog[1,2,5], Michelle Livne [6], Marcus Semtner[7,8], Tuan Leng Tay [9,10], Marco Prinz [11,12], Helmut Kettenmann[7,13], Matthias Endres [1,2,3,5,14,15,16], Golo Kronenberg[1,2,17], Ria Göttert [1,2,5,18] & Karen Gertz [1,2,3,5,18] ✉

Microglial proliferation is a principal element of the inflammatory response to brain ischemia. However, the precise proliferation dynamics, phenotype acquisition, and functional consequences of newly emerging microglia are not yet understood. Using multicolor fate mapping and computational methods, we here demonstrate that microglia exhibit polyclonal proliferation in the ischemic lesion of female mice. The peak number of clones occurs at 14 days, while the largest clones are observed at 4 weeks post-stroke. Whole-cell patch-clamp recordings of microglia reveal a homogeneous acute response to ischemia with a pattern of outward and inward currents that evolves over time. In the resolution phase, 8 weeks post-stroke, microglial cells within one clone share similar membrane properties, while neighboring microglia from different clones display more heterogeneous electrophysiological profiles. Super-resolution microscopy and live-cell imaging unmask various forms of cell-cell interactions between microglial cells from different clones. Overall, this study demonstrates the polyclonal proliferation of microglia after cerebral ischemia and suggests that clonality contributes to their functional heterogeneity. Thus, targeting clones with specific functional phenotypes may have potential for future therapeutic modulation of microglia after stroke.

Microglia, the main resident macrophages of the central nervous system (CNS), form a network of largely evenly distributed cells that continuously screen their surroundings and dynamically interact with their microenvironment[1]. In homeostatic conditions, random proliferation and apoptosis control the evenly distributed network of microglia[2]. In contrast, microglia respond to neuropathological conditions such as ischemic stroke[3], Alzheimer's disease (AD)[4], or traumatic CNS injury[5] with expanded proliferation, also known as microgliosis.

Several studies have described microgliosis after ischemic stroke[3,6–8]. However, the mechanism underlying this phenomenon remains elusive. We have previously shown, employing multicolor fate mapping, that microgliosis subsequent to facial nerve axotomy is driven by a shift in microglial proliferation from a stochastic process to selected clonal expansion[9]. It has further been demonstrated that clonal expansion of residual microglia is the underlying mechanism of microglia replenishment after pharmacological depletion[10,11]. More recently, it has been shown that microglial progenitors occupy the

niche of the developing brain through clonal expansion[12]. In all these previous studies, the proliferation dynamics involved the enlargement of many clones. Therefore, we opt here to use the term polyclonal proliferation to differentiate from cases where the term clonal expansion might indicate the domination of a single or very few clones, as classically observed in immunology and cancer biology[13,14].

In this study, we employ the multicolor fate mapping system *Cx3cr1^{creER/+}R26R^{Confetti/+}*−referred to as the Microfetti mouse[9]−to ascertain whether microgliosis subsequent to cerebral ischemia is driven by a clonal proliferation process of microglia similar to that observed in other conditions. We also aim to determine the long-term dynamics of this process after stroke in the transient model of 30-min middle cerebral artery occlusion (MCAo)/reperfusion. Recent transcriptomic studies have shown that microglial heterogeneity increases after ischemic stroke, with the emergence of multiple transcriptionally distinct clusters[15,16]. However, little is known about the factors underlying this diversity and whether clustered cells are clonally related. To address this question, we apply whole-cell patch-clamp analyses of microglia to explore whether clonal identity is one of the drivers of microglial functional heterogeneity after stroke. Additionally, we analyze the morphological spectrum of microglia after stroke and find parallels between the functional and morphological dynamics. Finally, we take advantage of multicolor labeling to study cell-cell interactions between neighboring microglia belonging to different clones and find distinct patterns of interactions. In summary, our current study establishes the dynamics of polyclonal proliferation of microglia after stroke and relates these dynamics to electrophysiological and morphological properties of microglia.

## Results

The Microfetti mouse is an established inducible system for multicolor fate mapping of microglia[9], created by crossing the *Cx3cr1^{CreER}* (refs. [17,18]) and the *R26R^{Confetti}* (ref. [19]) mouse lines. Upon tamoxifen injection and based on the location of the Cre recombinase activity, microglial cells randomly express one of four different fluorescent proteins (Supplementary Fig. 1a): membrane-tagged cyan fluorescent protein (mCFP), nuclear green fluorescent protein (nGFP), cytoplasmic yellow fluorescent protein (YFP), and cytoplasmic red fluorescent protein (RFP). The labeled cells maintain the expression of these reporters under a CAGG promoter and pass them onto their progeny upon cell division. CD11b+ CD115+ Ly6C^{lo} blood monocytes could also express the Confetti reporters. However, these cells have a half-life of 2-11 days[17,18] and were not detectable in blood 5 weeks after tamoxifen injection (Supplementary Fig. 1b). Taken together, the Microfetti mouse allows the tracing of microglia over time and space. It also enables the distinction between microglia and invading blood monocytes.

### The Microfetti (*Cx3cr1^{creER/+}R26R^{Confetti/+}*) mouse as a tool for studying microglial proliferation in ischemic stroke

We took advantage of this multicolor fate mapping system to study the dynamics of microglia proliferation in the 30-min proximal intraluminal MCAo/reperfusion model. We have chosen this model because of its high reproducibility, robust induction of microglia responses, and functional recovery, consequently, allowing the study of long-term sequelae of cerebral ischemia. This MCAo model mimics human large vessel occlusion with thrombectomy. The lesions are characterized by selective neuronal cell death limited to the ipsilateral striatum, resulting in a defined impairment of sensorimotor function[20]. After an intraperitoneal injection of tamoxifen and a waiting period of 6 weeks, mice were subjected to focal brain ischemia. Three days after stroke, MR-imaging was performed, which confirmed that all ischemic lesions included the dorsolateral striatum and showed homogeneous infarct sizes in the volume analysis (median = 10.26 mm$^3$ [IQR: 7.50−13.92]) (Supplementary Fig. 1c). The immunolabeled brain sections were analyzed at six different time points: 2 days, 1 week, 2 weeks, 4 weeks,

8 weeks, and 12 weeks after stroke (Fig. 1a). The ischemic region in the dorsolateral striatum and the corresponding region in the contralateral brain hemisphere were imaged using confocal laser scanning microscopy (Fig. 1b). As we have shown in our previous work, microglial cells in the non-ischemic hemisphere showed predominantly a classical ramified morphology[21] (Fig. 1c). In addition, the number of the Confetti+ cells in the non-ischemic contralateral striatum was stable over time with an average of 1653 ± 120 cells per mm$^3$, and a random distribution of the Confetti labeling among Iba-1+ cells (Fig. 1c, e). In contrast, Confetti+ microglia in the ischemic tissue showed substantial dynamic changes over time as regards the number of cells as well as their morphology and spatial distribution (Fig. 1d, e). At 2 days after stroke, Confetti+ microglia exhibited shorter ramifications and larger somata without a significant increase in their number in comparison to the contralateral side (3623 ± 639 vs. 2036 ± 427 per mm$^3$). After 1 week, the number of Confetti+ microglia on the ischemic side increased significantly in comparison to 2 days (11,368 ± 1966 vs. 3623 ± 639 per mm$^3$). Of note, many clusters of cells with a similar color appeared, indicating the formation of clones. These clones were prominent after 2 and 4 weeks without the domination of a single (or very few) clone(s) (Fig. 1d). From 2 weeks on, the number of Confetti+ microglia in the ischemic hemisphere decreased gradually. After 12 weeks, Confetti+ microglia returned to a level that was no longer statistically significant compared to the contralateral side (3425 ± 1503 vs. 1356 ± 386 per mm$^3$). Additionally, far fewer clones were apparent after 8 and 12 weeks.

### Microglia undergo polyclonal proliferation after stroke

To investigate whether the observed accumulation of Confetti+ microglia in ischemic tissue was indeed due to clonal proliferation dynamics, we applied a Monte Carlo simulation strategy as previously established[9,22]. After confocal microscopic image acquisition, 3D images were rendered to extract the location and the Confetti label of microglia in the ischemic region and the contralateral control region (Fig. 2a). Concentric rings, with radii ranging from 20 to 300 μm, were placed around each Confetti+ cell in a stack (Fig. 2b). The density of cells with the same color to the cell in question was calculated for each ring. Then, 1000 Monte Carlo simulations were applied for each stack, where the color labels of cells were randomized. Likewise, the density of cells with the same color in the concentric rings for the simulated data was calculated. The colored regions in Fig. 2b, c represent the areas between the 2nd and 98th percentiles of the Monte Carlo-simulated densities. If the recorded data is higher than the 98th percentile of the simulated data, it means that cells of the same color are closer to each other than they would be if labels were randomly distributed. In other words, it indicates a clonal relation of neighboring microglia. Our results suggest that clones start to appear as early as 2 days after ischemia (Fig. 2c). Clonality peaked after 2 weeks, (i.e., the highest difference between simulated and recorded data; Fig. 2c). Importantly, our data showed an enlargement in many distinct microglial clones indicating polyclonal proliferation dynamics (Fig. 1d and Fig. 2a). At subsequent time points, the recorded data gradually approached the simulated data to reach random levels after 12 weeks. This indicates a gradual re-establishment of the random distribution of labeled cells in the tissue and a disintegration of the clones. In contrast, the contralateral non-ischemic hemisphere showed a steady random distribution of labeled cells and no signs of clonal proliferation dynamics of microglia.

### The dynamics of polyclonal proliferation of microglia after 30-min MCAo

To quantify the spatiotemporal dynamics of the polyclonal proliferation, we followed a machine-learning approach. Specifically, we performed a cluster analysis using the density-based spatial clustering of applications with noise (DBSCAN) algorithm. Previous studies estimated the average nearest neighbor distance of microglia and

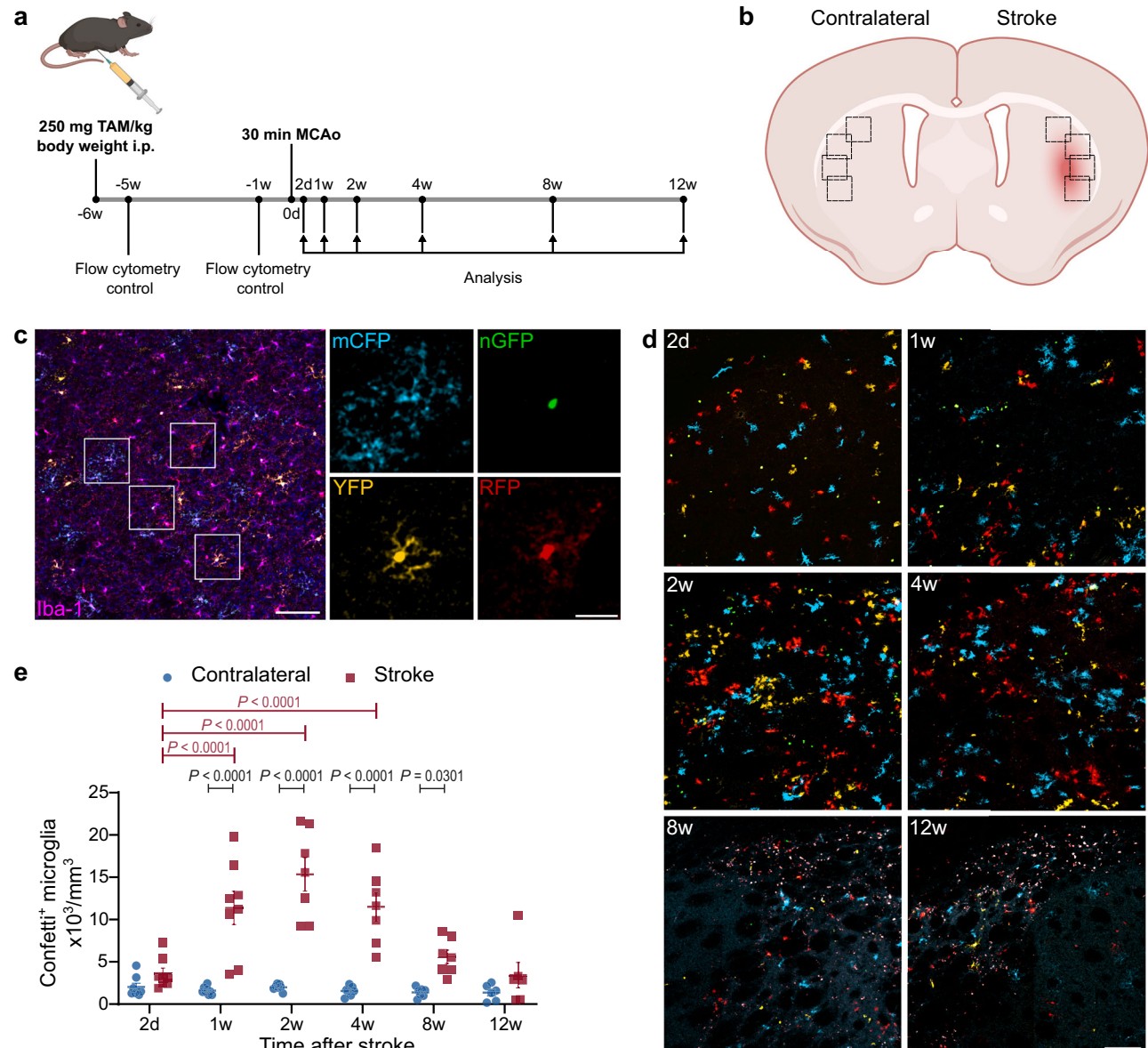

**Fig. 1 | Characterization of the *Cx3cr1^creER/+^R26R^Confetti/+^* (Microfetti) mouse model for fate mapping of microglia in ischemic stroke. a** Scheme for the experimental design of fate mapping of microglia after 30-min middle cerebral artery occlusion (MCAo) in Microfetti mice. Created in BioRender. Göttert, R. (2025) https://BioRender.com/rhlc1w4. **b** Illustration of the imaged regions of interest in the lateral part of the ischemic and contralateral striatum. For each animal, three consecutive brain slices were imaged with four images per slice per hemisphere. Created in BioRender. Göttert, R. (2025) https://BioRender.com/pzkky3d. **c** Representative fluorescent image of Iba-1⁺ microglia (magenta) in the contralateral striatum showing ramified morphology. In addition, many cells express one of four fluorescent proteins: membrane-tagged CFP (mCFP, cyan), nuclear GFP (nGFP, green), cytoplasmic YFP (yellow), and cytoplasmic RFP (red). Scale bars: 100 μm and 30 μm in the magnified images. **d** Representative images from the time course analysis of the ischemic lateral striatum from each investigated time point: 

2 days, 1, 2, 4, 8, and 12 weeks. After 1 week, there is a clear increase in the number of Confetti⁺ microglia, indicating microglial proliferation. In addition, the appearance of many clusters of cells expressing the same fluorescent marker and located close to each other suggests polyclonal proliferation dynamics. After 8 weeks, there is a clear decrease in the number of Confetti⁺ microglia, and large clones are less apparent. Lipofuscin autofluorescence signal is present at 8w and 12w time points. Scale bar: 100 μm. **e** Quantification of Confetti⁺ microglia in the ischemic and contralateral striatum showing the dynamics of microglial proliferation after 30-min MCAo. In contrast, the contralateral hemisphere displays a stable number of cells over the investigated period. Each dot represents one brain hemisphere from one mouse. Means ± s.e.m. are shown. Statistical analysis with two-way ANOVA, Šidák's multiple comparisons test for contralateral vs. stroke comparisons, and Dunnett's multiple comparisons test for time point comparisons. Number of animals $N = 8$ (2 d), 8 (1w), 7 (2w), 7 (4w), 7 (8w), 6 (12w).

identified 50 μm as a threshold for relatedness or clonality[9,11,12]. Hence, by specifying a radius of a neighborhood ($\varepsilon = 50$ μm) and a minimum number of cells in one clone (two cells), the DBSCAN algorithm clusters the cells based on their spatial density (Fig. 2d). Cells that express the same Confetti reporter and are closer to each other than 50 μm are considered to belong to the same clone. In contrast, cells with no neighbor closer than 50 μm are considered singlets, i.e., do not belong to any clone, and were excluded from further clonal analysis. This

allowed us to estimate the number of microglial clones and their size (number of cells per clone) over time (Fig. 2e, f). The number of clones increased abruptly during the first week after ischemia ($605 \pm 132$ vs. $2272 \pm 343$ per mm³ at 2 days and 1 week, respectively). After the first week, the number of clones stayed stable until 2 weeks. They decreased gradually afterward and were not statistically significant in comparison to the contralateral hemisphere after 12 weeks. When analyzing the average clone size dynamics, we found that it was about

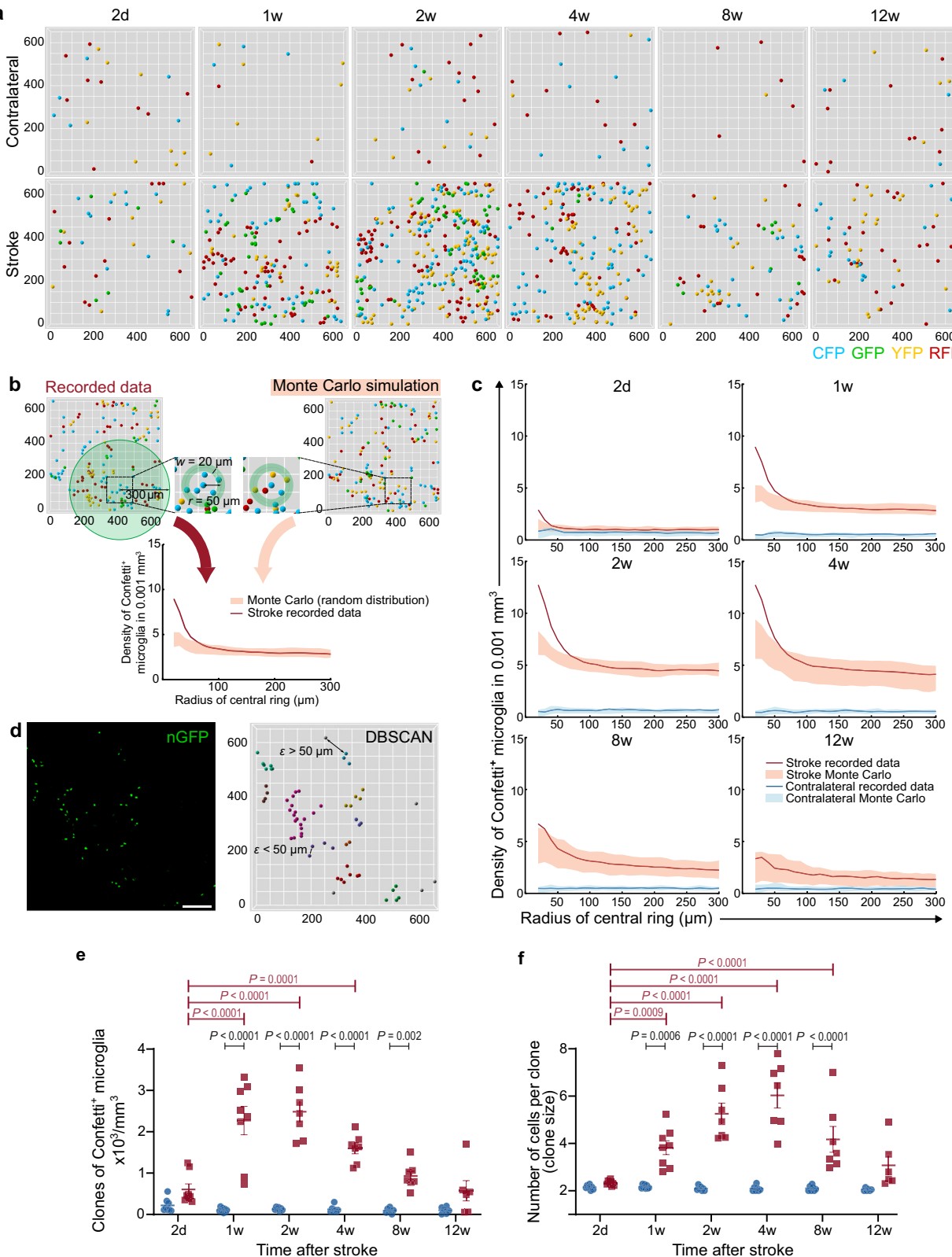

four cells per clone (3.8 ± 0.29) after 1 week, peaked after 4 weeks, reaching about six cells (6.03 ± 0.53), and decreased gradually afterward. In contrast, the contralateral side showed stable dynamics with mostly singlets and occasional small clones that are rarely larger than two cells. The distribution of clone sizes per animal in the stroke region and its contralateral counterpart are presented in Supplementary Fig. 2 and Supplementary Fig. 3. Additional proliferation dynamics were

investigated using the Ki-67 marker (Supplementary Fig. 4). The analysis revealed that the probability of microglia proliferation is highest 2 days after brain ischemia, with the majority of microglial proliferation occurring within the first 2 weeks after stroke (Supplementary Fig. 4b). The analysis also revealed an inversely proportional relationship between proliferation index and clone size (Supplementary Fig. 4c). To summarize, our data demonstrate the onset, peak, and

**Fig. 2 | Spatiotemporal dynamics of microglial polyclonal proliferation after ischemic stroke revealed by Monte Carlo simulation and machine learning.**
**a** Representative 3D renderings of Confetti[+] microglia in matched ischemic and contralateral striatum. Image dimensions are presented in μm. **b** Illustration of the applied Monte Carlo simulation method. Densities of same-colored cells were calculated within concentric rings with radii ranging from 20 to 300 μm and a fixed width of 20 μm. The enlarged image shows an example ring with a radius $r = 50$ μm. Hard lines represent recorded data, while colored regions represent randomized simulated data. The upper and lower bounds of simulated data correspond to the 98th and 2nd percentiles, respectively. **c** Time course analysis with Monte Carlo simulations for microglial post-stroke polyclonal proliferation. The polyclonal proliferation of microglia is present 2 days after stroke. The recorded cell densities reach their peak after 2 weeks and shift back toward random distribution after 12 weeks. The unaffected, corresponding region in the contralateral hemisphere shows no signs of clonal dynamics. **d** Cluster analysis with DBSCAN for an exemplary image from the GFP channel by employing a neighborhood radius ($\varepsilon$) of 50 μm. Cells that are further than 50 μm from their closest neighbor are considered singlets and were excluded from the clonal analysis in (**e**, **f**). Different colors stand for different clones. Scale bar: 100 μm. **e** The dynamics of the polyclonal proliferation of microglia represented by the number of clones of Confetti[+] microglia in ischemic and contralateral striatum for each investigated time point, calculated by DBSCAN. **f** The dynamics of changes in the clone size over time. In (**e**, **f**), each dot represents one brain hemisphere from one mouse. Means ± s.e.m are shown. Statistical analysis with two-way ANOVA, Šidák's multiple comparisons test for contralateral vs. stroke comparisons, and Dunnett's multiple comparisons test for time point comparisons. Number of animals (the same animals used in Fig. 1) $N = 8$ (2 d), 8 (1w), 7 (2w), 7 (4w), 7 (8w), and 6 (12w).

resolution dynamics of the local polyclonal proliferation of microglia after ischemic stroke.

## Microglia show distinct membrane properties over time after stroke

Microglia express a defined set of ion channels that are correlated to their functional state[23]. In the normal brain, microglial cells have a characteristic membrane current pattern, and this pattern changes in a time-dependent fashion after a pathological impact[24]. Therefore, to investigate the functional states of single microglial cells after stroke, we performed whole-cell patch-clamp recordings of microglia in acute Microfetti brain slices. We analyzed the membrane properties at 2 days, 1 week, and 8 weeks after stroke from microglial cells in the ischemic tissue and the corresponding contralateral region (Fig. 3a). The cells from the contralateral hemisphere from all time points were pooled in one control group. Membrane currents were recorded in the voltage clamp mode and voltage steps were applied for 50 ms ranging from −170 mV to +60 mV with 10 mV increment from a holding potential of either −70 mV (gray lines) or −20 mV (black lines) (Fig. 3b). In the contralateral hemisphere, microglia displayed a typical pattern of only small currents as previously described for ramified microglia in acute slices[24]. In contrast, microglial cells from the ischemic tissue after 2 days exhibited large inward currents at hyperpolarizing voltage steps and pronounced voltage-dependent outward currents that were activated after depolarization from a holding potential of −70 mV. Such currents were similar to those described in LPS-activated microglia in vitro[25] or in the facial nucleus in situ 24 h after facial nerve axotomy[24]. We have previously described these currents in invading monocytes 1 week after MCAo[21]. In line with our previous work, we observed inward currents after 1 week, while outward currents decreased[21]. Importantly, our patch-clamp analysis after 8 weeks detected pronounced inward currents; however, with a large variance between recorded cells. In accordance with the current-voltage relationships, microglia displayed a significantly decreased membrane resistance after 2 days that slowly but not completely recovered until 8 weeks after stroke (Fig. 3c). Microglial membrane capacitances were significantly increased only at 2 days after MCAo, indicating changes towards a rounder morphology and/or enlarged size (Fig. 3c). In contrast, the reversal potential did not change in the first week after MCAo but was significantly hyperpolarized after 8 weeks (median −66.56 mV). The analysis of the specific outward conductance revealed an increase for all time points after MCAo compared to control conditions. However, the most pronounced increase was found after 2 days. The specific inward conductance showed a progressive increase with time after MCAo, with significant changes after 1 and 8 weeks and a high variance between the cells at the latter time point. In conclusion, these results elucidate the electrophysiological profiles of microglia from the acute to the chronic phase after stroke and thereby reveal higher heterogeneity in membrane properties after 8 weeks.

## The properties of inward and outward currents of microglia

To investigate which types of ion channels are responsible for the observed electrophysiological profiles of microglia after stroke, we characterized the kinetics of the underlying currents. Notably, the outward current displayed the characteristics of voltage-dependent, delayed outward rectifying K[+] channels (Supplementary Fig. 5a). The time course of activation (tau) was voltage-dependent, with more rapid activation at more positive potentials. Currents did not inactivate within the 50-ms voltage step. The inward current activated with hyperpolarizing voltage steps at potentials negative to −100 mV showed the typical kinetics of inward rectifying K[+] channels (Supplementary Fig. 5b). The inactivation time constant decreased with more negative potentials. In addition, the analysis of the specific inward conductivity of cells with inward but no outward currents further differentiated between cells with moderate and cells with strong inward currents (threshold 0.5 nS/pF) (Fig. 3d).

Thus, by analyzing each cell for the presence of these currents, we found that microglial cells in our data could be classified into four groups: (a) cells with only small currents, (b) cells with moderate inward currents only, (c) cells with strong inward currents only, and (d) cells with inward and outward currents (Fig. 3e). Under control conditions in the contralateral hemisphere, 69% of cells displayed only small currents, while 31% showed moderate inward rectifying currents. At 2 days after stroke, 95% of microglia displayed both inward and outward currents. After 1 week, only 47% of cells displayed pronounced outward currents, while all cells retained their inward currents. Of note, 6% of cells showed strong inward current after 1 week. After 8 weeks, we observed a higher diversity with all four groups being represented. Interestingly, 12% of cells at 8 weeks exhibited only small currents, similar to control cells, while the group with strong inward currents increased to 24%. In addition, 28% of cells showed outward currents after 8 weeks that differed from the above-described voltage-dependent delayed outward-rectifying currents. Those currents were activated by depolarization without a consistent delay, showed a linear current-voltage curve, and were not inactivated at a holding potential of −20 mV. This indicates another ion channel at play 8 weeks after stroke (Supplementary Fig. 5c).

## Microglia within one clone share similar membrane properties, while neighboring microglia from different clones show higher heterogeneity 8 weeks after stroke

As described above, microglial cells within the ischemic tissue at 8 weeks after stroke showed high variability in their membrane properties and the types of membrane currents (Fig. 3e and Supplementary Fig. 5c). To investigate whether clonal identity contributes to this heterogeneity in microglial membrane properties, we compared recordings from cells within and between clones and between neighboring cells that were <50 μm apart but had different Confetti markers (Fig. 4a). The distance parameter of 50 μm ensures that patched

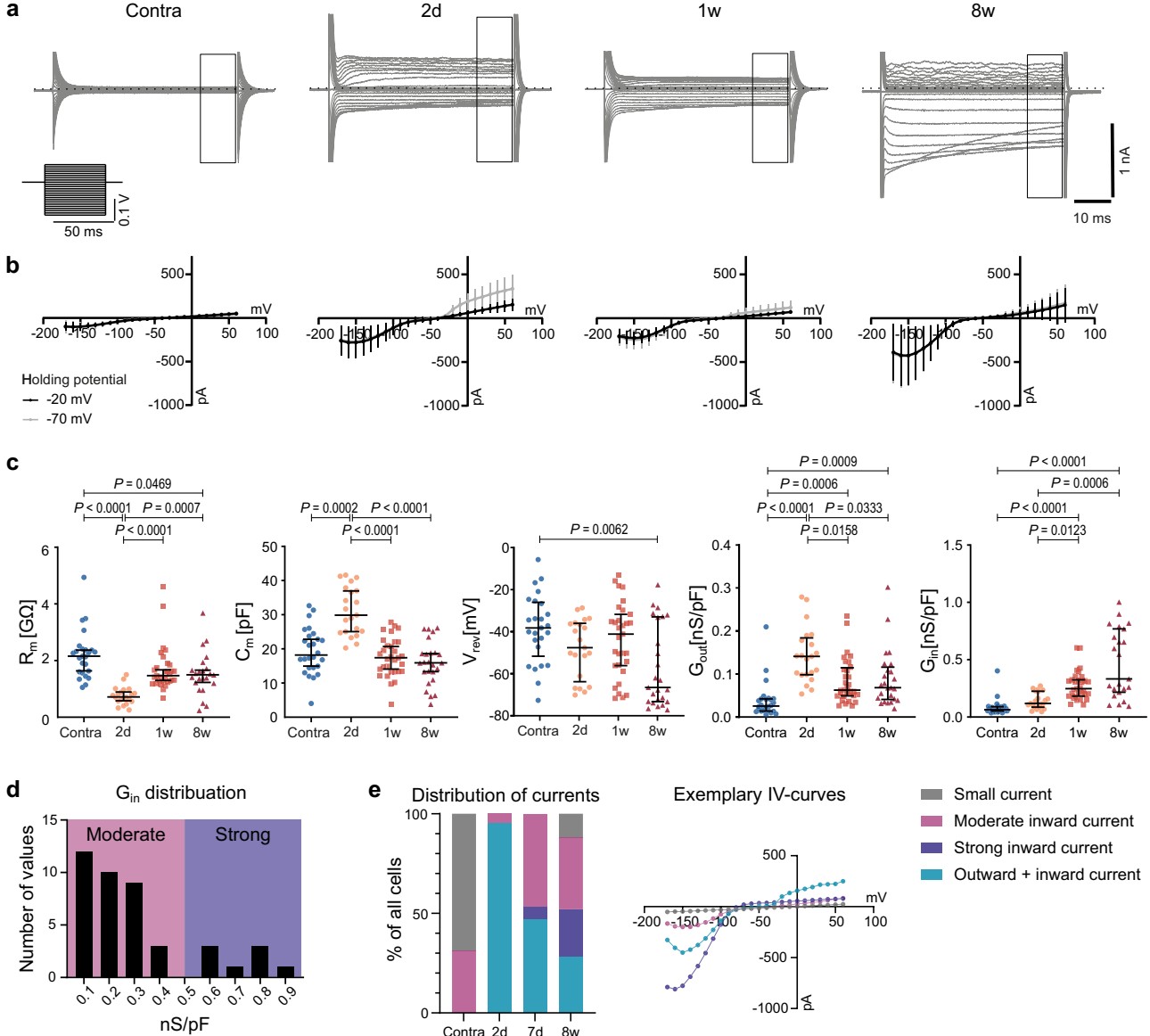

**Fig. 3 | Microglia membrane properties change over time after stroke. a** Whole-cell patch-clamp experiments on microglia 2 days, 1 week, and 8 weeks after 30-min MCAo are shown. Control cells were examined in the contralateral uninjured hemisphere (Contra). Exemplary traces recorded at a holding potential of −70 mV. Voltage steps from −170 mV to +60 mV for 50 ms each (schematic on the left). **b** Corresponding IV-curves at −20 mV holding potential (black) and −70 mV (gray). Data are represented as mean ± standard deviation. Note the increasing variance after 8 weeks. Number of cells $n$ (−20 mV, −70 mV) = 26, 25 (Contra); 21, 21 (2 d); 31, 31 (1w); 25, 24 (8w). **c** Calculation of membrane resistance (holding potential −20 mV), capacitance (holding potential −20 mV), and reversal potential (holding potential −70 mV) derived from the IV-curves. Specific outward conductance derived from steps at 0 mV and −20 mV, 40 ms after the beginning of pulse, and normalized to each cell's capacitance (holding potential −70 mV). Specific inward conductance derived from steps at −120 mV and −100 mV, 10 ms after the

beginning of pulse, and normalized to each cell's capacitance (holding potential −20 mV), showing high variance at 8 weeks. Data are represented as median and interquartile range. Statistical analysis with Kruskal-Wallis test. $n$ = 26 (Contra) for $C_m$, $G_{out}$, and $G_{in}$, 25 for $R_m$ and $V_{rev}$; 21 (2 d); 31 (1w); 25 (8w) for $C_m$, $V_{rev}$, and $G_{out}$, 23 for $R_m$ and $G_{in}$. **d** Histogram plotted for specific inward conductance (−120 mV to −100 mV, 10 ms after stimulus onset, holding potential −20 mV) for cells with inward but no outward current. This provides the basis for differentiating the two groups of cells with moderate and cells with strong inward currents (threshold 0.5 nS/pF). **e** Distribution of inward and outward currents over different time points with exemplary IV-curves (holding potential −70 mV) showing four different types of cells based on the types of displayed currents. Nine cells under control conditions showed moderate inward currents. Please note the homogeneous response at 2 days and the increased heterogeneity over time.

neighboring cells from different clones share a similar microenvironment, as is the case for patched cells within a clone. Cells from the same clone exhibited very similar membrane properties and currents (Fig. 4b). In fact, the high similarity in electrical properties of cells within one clone was also evident at 1 week after stroke (Supplementary Fig. 6a). In contrast, neighboring cells from different clones are less similar in their electrophysiological profiles (Fig. 4c and Supplementary Fig. 6b).

## Revealing the morphological diversity of microglia after stroke using deep learning

Next, we investigated whether microglia exhibit morphological changes that align with the observed electrophysiological dynamics. High-magnification images were acquired of RFP⁺ microglia in the ischemic striatum and the contralateral hemisphere (Supplementary Fig. 7a). To obtain an accurate image segmentation for such diverse morphologies, we developed an automated pipeline for

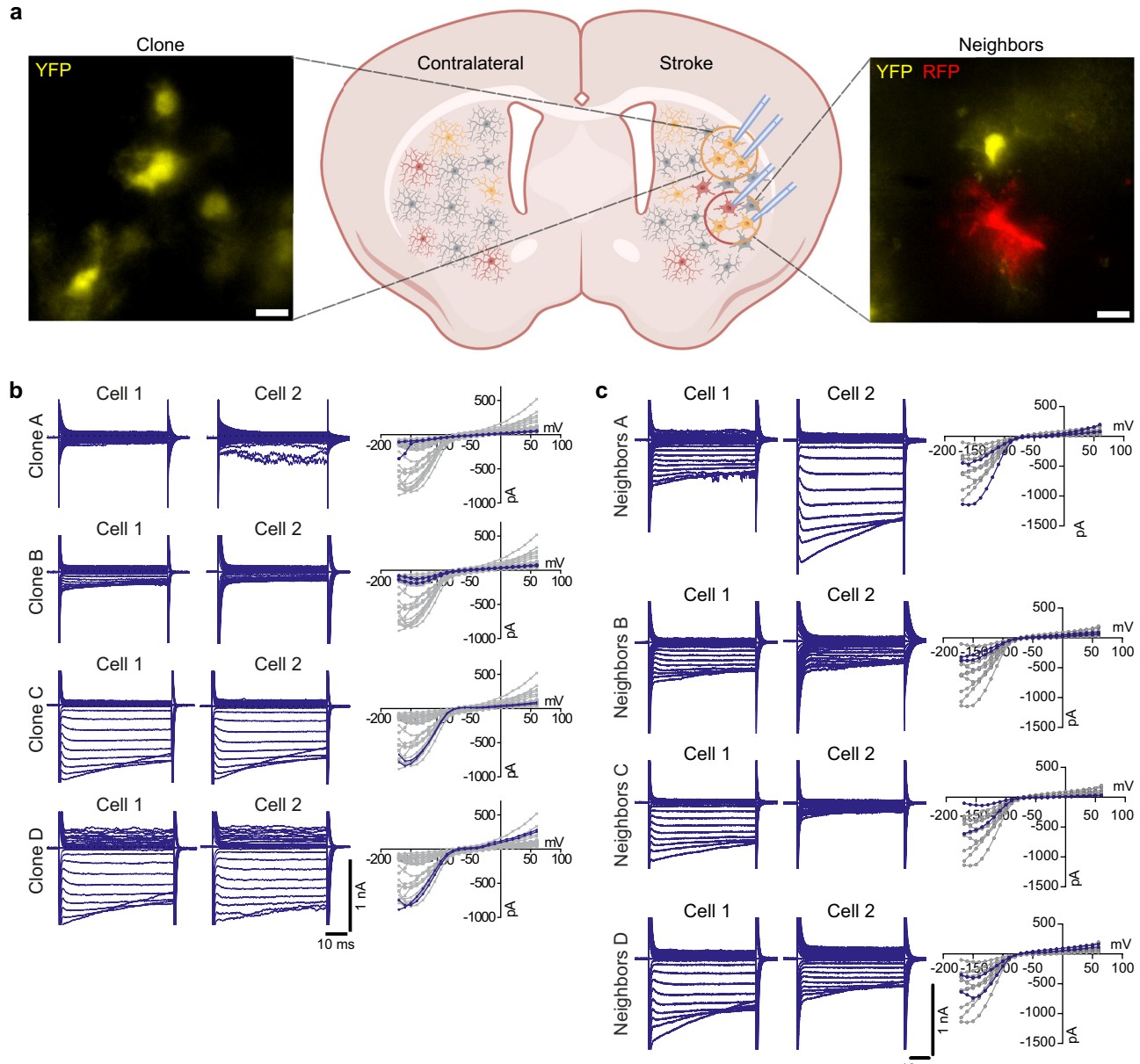

**Fig. 4 | Microglial cells within one clone are more functionally correlated than neighboring cells from different clones 8 weeks after MCAo. a** Scheme of two whole-cell patch-clamp experiments 8 weeks after MCAo. In the first experiment, multiple cells within one clone were patched. In the second experiment, neighboring cells (distance <50 μm) expressing different Confetti labels were patched. In these experiments, YFP and RFP signals were used to visualize microglia and identify clonal identity. The left microscopic image shows an example of a YFP+ clone in the ischemic tissue 8 weeks after stroke, while the right image shows two neighboring cells from different clones. Scale bar: 10 μm. Created in BioRender. Göttert, R. (2025) https://BioRender.com/m54j491. **b** Cells from the same clone show similar currents. Each row represents raw traces and IV-curves from two different cells from one clone. The IV-curves show the presented traces in blue against all other cells at the time point in gray. Despite the increased overall heterogeneity after 8 weeks (Fig. 3e and Supplementary Fig. 5c) and the large differences between clones, cells of the same clone display very similar currents. **c** The membrane properties of neighboring cells from different clones are less correlated than those of cells from the same clone. Neighbors A and C show very distinct membrane properties, while neighbors B and D are more similar. Additional recordings of neighboring cells are shown in Supplementary Fig. 6b.

morphological analysis, which employs a convolutional neural network U-Net[26] (Supplementary Fig. 7b). After training the U-Net model, 668 cells were analyzed, and multiple morphological features were extracted for each cell. Some features distinctly differentiated between ipsi- and contralateral microglia and between cells from the ischemic tissue at different time points after stroke (Fig. 5b). Specifically, microglia in the stroke region showed an abrupt decrease in surface area-to-volume ratio at 2 days and a gradual recovery afterward. These changes are congruent with the observed increase in cell capacitance of microglia at 2 days, which also correlated with the

presence of the delayed outward rectifying current described in Supplementary Fig. 5a. Based on the morphological feature presented in Fig. 5b, a uniform manifold approximation and projection for dimension reduction (UMAP) analysis was performed. The analysis illustrates an evident distinction between highly ramified cells, predominantly made of cells from the contralateral hemisphere, and cells in the stroke region (Fig. 5c and Supplementary Fig. 7c). It also shows a clear polarity of microglial morphologies in the stroke region where enlarged (primed) cells, mostly present at 2 days and 1 week after stroke, are localized at the upper right pole of the graph. In contrast,

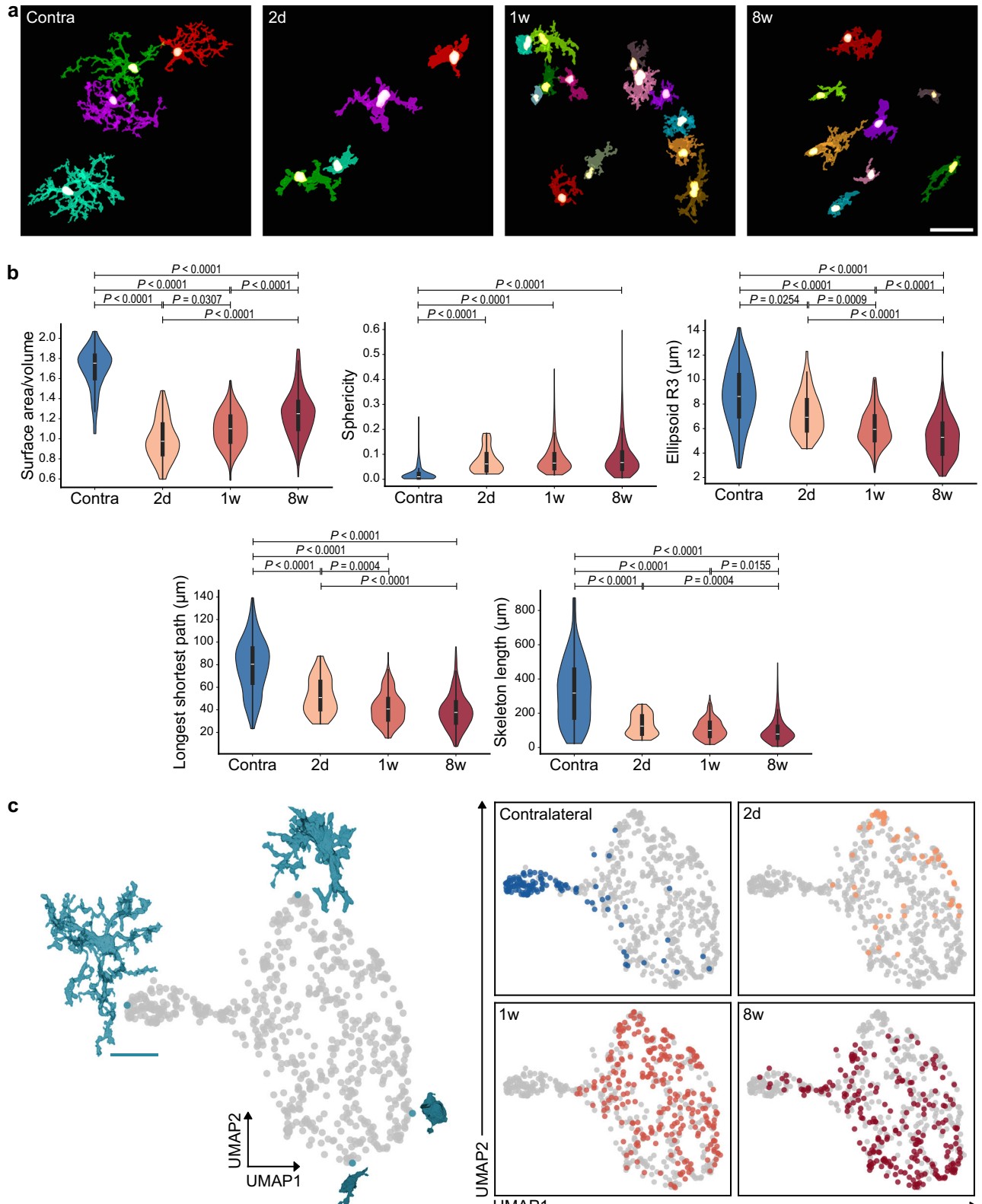

small roundish (ameboid) cells are more present at 8 weeks and occupy the lower right pole of the graph. Interestingly, highly ramified cells were absent in the ischemic tissue at 2 days and 1 week after stroke but reappeared at 8 weeks. Such morphological recovery after 8 weeks supports the observation of electrophysiological recovery of some microglial cells, as shown in Fig. 3e.

**Microglial inter-clonal cell-cell interactions after stroke**

In the morphological analysis, we observed microglial cells in the ischemic tissue in close contact with each other (Fig. 5a and Supplementary Fig. 8a). Therefore, we sought to exploit the multicolor labeling of microglia in the Microfetti mouse to visualize the microscopic interactions between microglial cells in the ischemic tissue.

**Fig. 5 | Morphological heterogeneity of microglia after ischemic stroke.**
**a** Representative 3D renderings of RFP⁺ microglia from the contralateral and
ischemic striatum 2 days, 1 week, and 8 weeks after stroke. Each cell is displayed in a
distinct color. Scale bar: 30 μm. **b** Violin plots for selected morphological para-
meters of microglia, i.e., surface area/volume, sphericity, ellipsoid radius 3 (the
length of the smallest semi-axis of the fitted inertia ellipsoid), the longest shortest
path, and the total skeleton length. 5 images per brain hemisphere were collected
from 5 animals at each time point. Images from the contralateral hemisphere of all
mice were pooled into one group. The violin outlines represent the kernel density
estimation of the data distribution, limited to the observed minima and maxima.
The embedded boxplot shows the median (central line), the interquartile range
(IQR; box bounds at the 25th and 75th percentiles), and whiskers extending to the
most extreme values within 1.5 × IQR from the quartiles. Statistical analysis using
Kruskal-Wallis test followed by Dunn's test for multiple comparisons. Number of

cells $n$ = 668, 123 (Contra), 58 (2 d), 256 (1w), 231 (8w). Please note that variations in
cell numbers are related to proliferation dynamics. **c** UMAP plots based on the five
selected morphological parameters of 668 microglia shown in gray. The visualized
cells represent the minimum and maximum values of the UMAP components
(UMAP1 and UMAP2). Scale bar: 20 μm. The left pole of the graph represents a
distinct cluster of highly ramified cells belonging predominantly to the con-
tralateral striatum. Interestingly, the highly ramified cells are absent at 2 days and
1 week but reappear at 8 weeks. Cells from the ischemic tissue cluster to the right,
with enlarged primed cells from 2 days and 1 week occupying the upper right pole
of the graph and smaller, more spherical cells occupying the lower right pole of the
graph. Interestingly, this morphological gradient, which is shown in more detail in
Supplementary Fig. 7c, corresponds well to how cell morphology evolves over time
as presented here.

Indeed, when we investigated the ischemic region, we observed cells
from different clones in direct interaction with each other. For exam-
ple, what appears as a single object in Iba-1 immunofluorescence in
Fig. 6a is composed of two microglial cells from two different clones.
High-resolution confocal images and their 3D renderings showed how
the two cells wrapped around and interacted with each other (Fig. 6a
and Supplementary Movie 1). In fact, we could differentiate between
four different types of cell-cell interactions of microglia in the ischemic
tissue (Supplementary Fig. 8b): process-soma, process-process, soma-
soma (flat), and entangled soma-soma interactions. The presence of
these interaction types was also confirmed with stimulated emission
depletion (STED) super-resolution microscopy (Fig. 6b). To explore
the temporal dynamics of these interactions, we also employed live-
cell imaging on acute brain slices. Indeed, we observed process-to-
soma interactions, where a microglial cell extends a process to touch
the soma of an adjacent microglial cell (Fig. 6c and Supplementary
Movie 2). We also observed process-to-process interactions, where a
microglial cell connects with one or more neighboring microglial cells
through their processes. Supplementary Fig. 9a and Supplementary
Movie 3 show a YFP⁺ process dynamically sliding over another process
from an RFP⁺ cell. Furthermore, microglial cells showed soma-to-soma
interactions. In some cases, the two somata were located next to each
other in a parallel or flat manner (Supplementary Fig. 9b and Supple-
mentary Movie 4). In other cases, the two or more microglial cells
appeared entangled together in a nodular formation (Supplementary
Fig. 9c and Supplementary Movie 5). It is important to emphasize here
that those inter-clonal interactions are not due to cell proliferation
because, in a proliferation event, both microglial cells share the same
labeling color. Since microglia in homeostatic conditions show mini-
mal territorial overlap[27], a similar analysis in the contralateral hemi-
sphere is of little relevance.

## Discussion

The present study provides several findings to advance the current
understanding of microglial responses to ischemic stroke. First, we
show that microglia undergo polyclonal proliferation after ischemic
stroke and characterize the dynamics of this process. Second, we
demonstrate the heterogeneity in the electrophysiological and mor-
phological profiles of microglia over time after stroke and describe a
partial functional and morphological recovery after 8 weeks. Third, our
data shows a functional correlation of microglial cells within a given
clone and a larger difference between neighboring cells from different
clones in the resolution phase, which highlights the impact of clonal
identity on microglial function post-stroke. Fourth, we demonstrate
inter-clonal cell-cell interactions of microglia after stroke.

Microgliosis after ischemic stroke has been investigated using
unicolor reporter mouse models such as *Cx3cr1*<sup>GFP</sup> or immunohisto-
chemical labeling with pan-macrophage markers such as Iba-1, F4/80,
or isolectin B4[3,6–8]. While these methodologies can be employed to
quantify the number of microglial cells, they are inadequate for

elucidating the underlying clonal dynamics of this process. In this
study, multicolor fate mapping was employed to demonstrate that
microglia transition from a stochastic mode of proliferation under
homeostatic conditions to polyclonal proliferation upon stroke
induction. Our electrophysiological experiments suggest that clonality
might have an impact on the functional heterogeneity of microglia
after brain ischemia. Indeed, even within a distance of 50 μm, micro-
glial neighbors from different clones exhibited more distinct electro-
physiological profiles than neighboring microglia sharing the same
Confetti color or clone. Single-cell RNA sequencing studies on
experimental stroke models have yielded evidence of the hetero-
geneity of microglia after stroke at a transcriptomic level[15,16]. The
present study provides supportive evidence for such heterogeneity on
a functional level as well.

Several preclinical studies have sought to target microglia or their
proliferation following a stroke. In general, most studies have shown
that microglia depletion or a pan-inhibition of their proliferation
worsens the effects of stroke[28–31]. However, it has also been observed
that microglia depletion may improve stroke outcome[32]. Indeed, there
is accumulating evidence that microglia play a complex role in stroke
pathophysiology, contributing to both inflammatory and neuropro-
tective mechanisms[33]. A recent study, for example, has shown that a
microglia-specific HDAC3 knockout specifically inhibits the prolifera-
tion of proinflammatory microglia, thus improving long-term func-
tional and histological outcomes after stroke[34]. Given the functional
heterogeneity among microglial clones, our results support the notion
that future studies should employ such differential proliferation-
related reprogramming to specifically promote the proliferation of
neuroprotective clones.

In this study, we used whole-cell patch-clamp to investigate
functional alterations in microglia at the single-cell level. Prior research
has established a link between the functional state of microglial cells
and the specific membrane currents they exhibit[24,35–37]. Here, we
observed characteristic dynamics of a delayed outward rectifier, such
as Kv1.3, a channel associated with stroke pathology[38]. Kv1.3 has been
linked to an activated proinflammatory microglial phenotype, and its
pharmacological inhibition has demonstrated protective effects in
stroke and other neuroinflammatory conditions[38,39]. Conversely, the
kinetics of inward rectifying currents align with inward rectifying
potassium channels, such as Kir2.1, which has been associated with
microglial migration[37] and IL-4 stimulation in vitro[40,41].

Our study also investigates the proliferation dynamics of clonally-
derived microglia and their electrophysiological and morphological
profiles over an extended period of time. Our Monte Carlo simulations
and Ki-67 analysis show that microglia start to clonally proliferate as
early as 2 days after MCAo. This process is accompanied by a sig-
nificant decrease in surface area-to-volume ratio and skeleton length
and a significant increase in membrane capacitance. At this stage,
microglial cells exhibited a relatively uniform electrophysiological
profile, characterized by the presence of both inward and outward

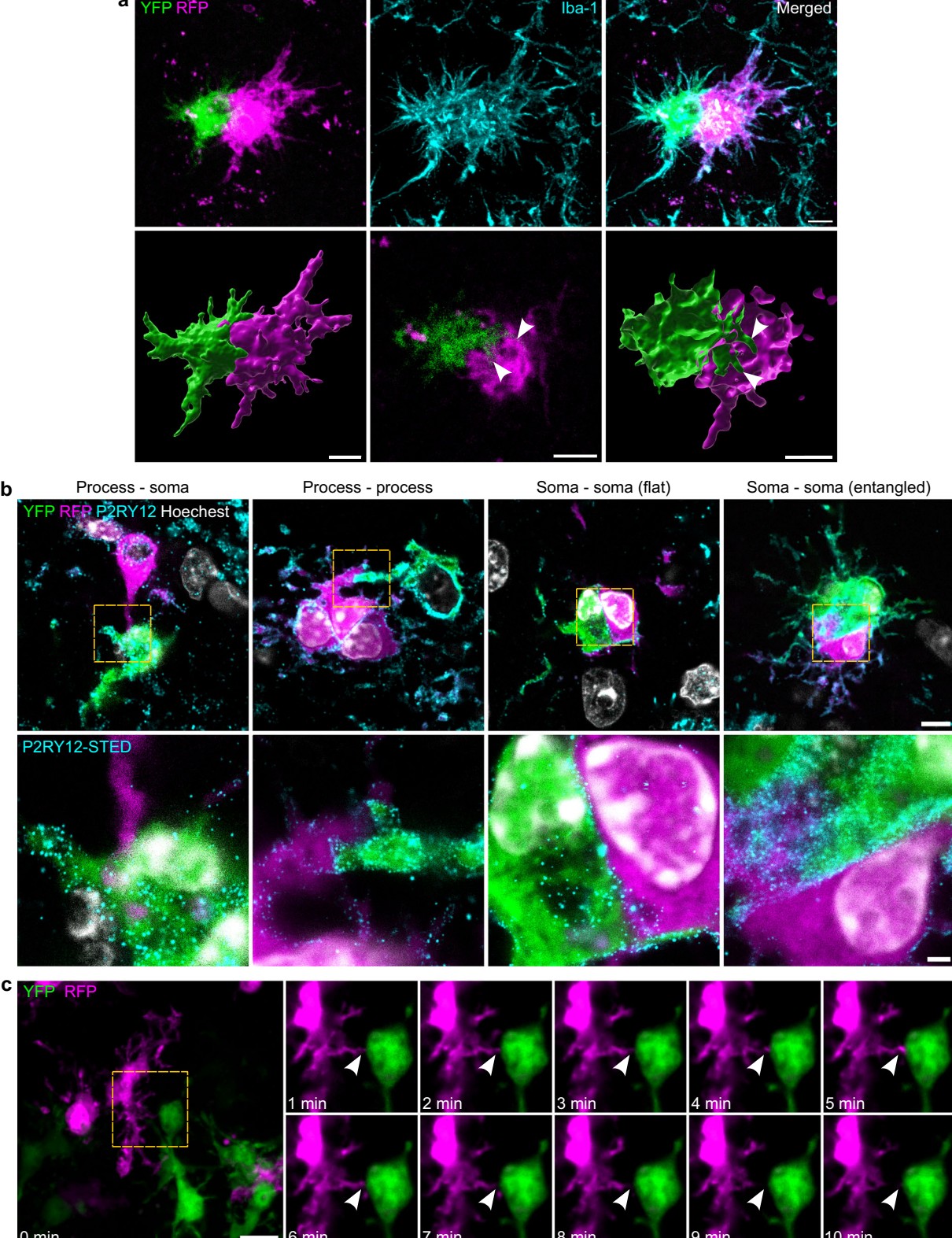

**Fig. 6 | Microglial inter-clonal cell-cell interactions after ischemic stroke. a** YFP⁺ (in green) and RFP⁺ (in magenta) microglial cells in close proximity. The upper row shows a maximum intensity projection (MIP) of a *z*-stack illustrating the entangled two cells. The borders between the two cells are obscured in the Iba-1 channel (cyan). The lower row shows a 3D rendering, a single *z*-plane, and a sectioned 3D rendering visualizing how the RFP⁺ cell wraps around protrusions from the YFP⁺ cell (white arrowheads) (see Supplementary Movie 1). Scale bars: 5 μm. **b** STED microscopy images showing different types of microglial cell-cell interactions. All images represent a single *z*-plane. The lower row shows high-resolution images with the P2RY12 channel (cyan) in STED. Scale bar: 5 μm and 1 μm for the upper and lower row, respectively. **c** Live-cell imaging of acute brain slices for 10 min reveals dynamic inter-clonal cell-cell interactions. The left image is an MIP, while the enlarged images depict a single *z*-plane over a 10-min recording period. The RFP⁺ cell extends a process (white arrowhead) to establish contact with the soma of a YFP⁺ cell and retracts it later (see Supplementary Movie 2). Scale bars: 10 μm for the left image and 3 μm for the enlarged images.

currents. After 1 week, we have observed a drastic increase in the number of clones with higher electrophysiological and morphological diversity, as the presence of the delayed outward rectifying current decreased. Microglial polyclonal proliferation reached its peak 2 weeks after MCAo, with a small percentage of cells undergoing proliferation at that time point. Afterward, there was a gradual decrease in the number of clones but an increase in the heterogeneity of electrophysiological profiles after 8 weeks. Specifically, some microglial cells retained the electrophysiological states observed at 1 week with increased inward currents, while others transitioned towards different activation states with altered currents, and a subset displayed a profile similar to control cells found in homeostatic tissue. This ability of microglia to recover after ischemia is further supported by our morphological analysis, which revealed the emergence of highly ramified microglial cells 8 weeks post-stroke. Such electrophysiological recovery was previously noted in other disease models, such as facial nerve axotomy and stab wounds[24,36]. These findings underscore the dynamic and adaptable nature of microglial responses to ischemia over an extended time frame.

Another finding of our study is the discovery of inter-clonal cell-cell interactions among microglia within the ischemic tissue. Previous stroke research has predominantly concentrated on microglial interactions with either neurons[28,42] or other immune cells[21,43–46]. However, little attention has been devoted to understanding the intricate dynamics of how microglia engage and harmonize their responses to ischemic conditions within their network. Notably, a comprehensive study by Scheiblich et al.[47] has demonstrated the capacity of microglia to build adaptive and on-demand functional networks for degrading α-synuclein fibrils, both in vitro and in vivo. These intricate networks emerge as microglia establish intercellular connections, facilitating the transfer of α-synuclein and mitochondria between cells. Our observations in fixed slices and ex vivo live-cell imaging in stroke tissue highlight that microglia can interact with neighboring microglial cells that originate from different clonal origins. Consequently, such intercellular connections are not mere remnants of incomplete cell separation during mitosis but rather a proactive process of interconnection between unrelated microglial cells. It is worth mentioning that we did not observe any flow of fluorescent proteins between contacting cells, which would argue against a free exchange of cytoplasmic material.

Moreover, our observations show that microglia exhibit the capacity to establish direct soma-to-soma interactions, which is a significant departure from their usual territorial behavior under homeostatic conditions. In fact, such behavior has been previously reported using electron microscopy in an experimental stroke model[48]. Another previously described form of microglial interaction is nodule formation in the context of multiple sclerosis and stroke pathology in humans[49,50]. Indeed, recent research has indicated that white matter-associated microglia form such nodules in a TREM2-dependent manner as they engage in clearing degraded myelin during aging[51]. Our findings illustrate different styles of microscopic proactive interactions between microglia. The implications of those different connection styles in the context of stroke and other disease models remain to be fully explored. Future studies should aim to understand the underlying molecular pathways and elucidate the functional advantages that microglia derive from such behavior.

When interpreting our findings, a few technical aspects and limitations need to be considered. First, the Microfetti mouse model allows us to exclude the contribution of invading monocytes. However, $Cx3cr1^+$ CNS-associated macrophages (CAM) could also be labeled as we have shown previously[52,53]. Our study focused on brain parenchyma, but perivascular macrophages could contribute to the observed effects. Multicolor fate mapping systems that differentiate CAM from microglia are emerging and could be used in the future[53]. Second, in a four-color system, it cannot be completely ruled out that close neighboring cells might acquire the same color by chance upon tamoxifen injection. To minimize this possibility, we used a tamoxifen dosage that results in sparse labeling of microglia. Indeed, the images from the nonischemic contralateral striatum show this probability to be less than 10% in most animals (Supplementary Fig. 2). RGB-cell labeling systems offer a broader spectrum of colors and enhance the depth of clonal analysis[12,54]. However, the currently available systems also present disadvantages, such as off-target labeling and invasive intracranial stereotactic injections. Future studies would benefit from the development of noninvasive, microglia-specific, combinatorial fluorescent protein fate mapping systems[55]. Third, the dynamics we describe may differ in various stroke models or aged mice. The 30-min MCAo model is ideal for studying microglial proliferation without significant glial loss[21,56,57], but longer occlusion times or permanent occlusion may result in different dynamics[58].

In summary, microglia play a critical role in the inflammatory response to ischemia, and their modulation has been explored as a potential therapeutic approach for stroke[59]. However, previous efforts to target microglia for stroke treatment have faced challenges in translating to clinical success[60]. This underscores the significance of gaining a comprehensive understanding of how microglia respond to stroke, particularly the intricacies of their dynamic behavior. The present study offers valuable insights into these dynamics and proposes them as potential targets for future interventional studies.

## Methods

### Mice
All experimental procedures were approved by the State Office for Health and Social Affairs in Berlin (LAGeSo) and conducted following the German Animal Welfare Act and Animal Welfare Regulation Governing Experimental Animals (TierSchVersV) under the approved animal protocol with registration number G 0057/20. Microfetti ($Cx3cr1^{creER/+}R26R^{Confetti/+}$) mice were generated as described and characterized earlier[9]. All experiments were conducted on young adult female mice aged 7–13 weeks at the time of tamoxifen injection. Genotyping for $Cx3cr1^{creER}$ and $R26R^{Confetti}$ was performed according to Jackson Laboratory stock no. 020940 and stock no. 013731, respectively. Mice were group housed with *ad libitum* access to food and water and a 12 h light/dark cycle.

### Administration of Tamoxifen
Tamoxifen solution was prepared by dissolving tamoxifen (Sigma-Aldrich, T5648-1G) in corn oil (Sigma-Aldrich, C8267-500ML) with a concentration of 20 mg/ml. Six weeks prior to stroke induction, mice received a single i.p. injection of tamoxifen (250 mg/kg body weight), e.g., a mouse weighing 20 g received 5 mg tamoxifen.

### Blood collection and flow cytometry
Blood was collected from the facial vein in capillary blood collection tubes (Microvette® 500 K3 EDTA, 20.1341.100). Within 30 min, it was transferred into 5 mL polystyrene tubes (Falcon®, 352235) and incubated with rat anti-Mouse CD16/CD32 antibody (BD Biosciences, 553141) for 5 min at 4 °C. The following antibodies were added, and the samples were incubated for 30 min at 4 °C: CD45-BV711 (BioLegend, 109847), CD11b-APC (BioLegend, 101212), CD115-APC-Cy7 (BioLegend, 135531), Ly6C-PE-Cy7 (BioLegend, 128018). Afterward, we lysed the red blood cells with BD Pharm Lyse (BD Biosciences, 555899) according to the manufacturer's instructions. The samples were washed twice and resuspended with FACS buffer (PBS, 0.5% BSA) and analyzed with BD® LSR II Flow Cytometer. Data were analyzed with the FlowJo software v10.6.1.

### Induction of cerebral ischemia
Cerebral ischemia was induced by following a well-established and standardized protocol in our lab, as described elsewhere[21,61]. In brief,

anesthesia was induced by 1.5–2% isoflurane and maintained during the surgery by 1.0% isoflurane in 70% $N_2O$ and 30% $O_2$ using a vaporizer. The left MCA was occluded by introducing a 7–0 silicone rubber-coated monofilament (Doccol Corporation, 7019910PK5Re) into the internal carotid artery. The filament was advanced until the A1 segment of the anterior cerebral artery, thus occluding the origin of the MCA. After an occlusion time of 30 min, the animals were re-anesthetized, and the filament was removed to allow reperfusion. This procedure results in an ischemic lesion mostly confined to the dorsolateral caudoputamen[62]. All further analyses were performed in that region at a Bregma level from +0.145 to +0.545, which is certainly involved after a successful MCAo surgery.

## Magnetic resonance imaging (MRI)

MRI scans were conducted 3 days after MCAo to ensure a successful induction of cerebral ischemia. The measurements were performed with a 7 Tesla rodent scanner, Pharmascan 70/16 (Bruker BioSpin), and a 20-mm-1H-RF quadrature-volume resonator. Anesthesia was induced with 1.5–2% isoflurane and maintained at 1.0% isoflurane in 70% $N_2O$ and 30% $O_2$ during the measurement. The respiration rate was constantly monitored with an MRI-compatible monitoring and gating system for small animals (SA Instruments, Inc.). A T2-weighted 2D turbo spin-echo sequence was used (imaging parameters TR/TE = 4200/36 ms, rare factor 8, 4 averages, 32 axial slices with a slice thickness of 0.5 mm, field of view of 2.56 × 2.56 cm, matrix size 256 × 256). The lesion volume was quantified using the software Analyze v10.0 (AnalyzeDirect).

## Preparation of acute slices

Acute slices were prepared from adult Microfetti mice, 2 days, 7 days, and 8 weeks after MCAo for patch-clamp experiments and 8 weeks after MCAo for live-cell imaging. Slice preparation was performed as previously described[63]. Mice were decapitated, the cranium was opened, and the brain was carefully transferred to ice-cold preparation solution (230 mM Sucrose, 26 mM $NaHCO_3$, 2.5 mM KCl, 1.25 mM $NaH_2PO_4$, 10 mM $MgSO_4$, 0.5 mM $CaCl_2$, 10 mM Glucose, pH 7.4). The cerebellum and the brain stem were separated from the cerebrum. The cerebrum was then glued to a cold metallic disc and cut into 250 μm-thick coronal slices using a Microm HM 650 vibratome (Microm International GmbH). Slices of the striatum were transferred to oxygenated aCSF (134 mM NaCl, 2.5 mM KCl, 1.3 mM $MgCl_2$, 2 mM $CaCl_2$, 1.26 mM $K_2HPO_4$, 10 mM Glucose, 26 mM $NaHCO_3$, pH 7.4, osmolarity 340 mmol/kg; gassed with 95% $O_2$, 5% $CO_2$) and stored at room temperature for approximately 30 min before electrophysiological recordings or live-cell imaging started.

## Patch-clamp recordings

Whole-cell patch-clamp recordings were performed in a custom-made chamber in submerged conditions. Pipettes were pulled from borosilicate capillaries using a Sutter Instrument Pipette Puller (HEKA Elektronik) and filled with standard intracellular solution (120 mM KCl, 10 mM EGTA, 25 mM HEPES, 5 mM NaCl, pH 7.2, osmolarity 310 mmol/kg). Pipettes were mounted on a headstage and connected to a HEKA double patch-clamp EPC10 amplifier (HEKA Elektronik). Slices were superfused with oxygenated aCSF at room temperature. Cells were visualized using an Axioskop 2 FS plus microscope (Zeiss) with an immersive 63x objective and a Photonics LPS-150 fluorescent lamp (Photonics). YFP and RFP signals were utilized to identify microglia because the cytoplasmic distribution of the two reporter proteins allows adequate visualization of the cell and its borders, facilitating efficient patching. Clones were identified as microglial cells expressing one color (YFP or RFP) and having a distance of approximately 50 μm from the nearest neighbor. Neighboring cells of different clones were identified as cells expressing different colors and being less than 50 μm apart. YFP was excited at 480 nm, emitting light at 521–561 nm; RFP

was excited at 550 nm, emitting light at 580–630 nm. Membrane currents were recorded at a holding potential of −20 mV and −70 mV with de-/hyperpolarizing steps (from −170 mV to +60 mV) of 50 ms duration. Capacitive transients of the pipette were compensated by TIDA 5.24 software (HEKA Elektronik). Further analysis of electrophysiological data was performed using custom-written scripts in Igor Pro 6.37 (WaveMetrics).

## Live-cell imaging

Acute brain slices were transferred into a 35-μm microscopy dish equipped with a 1.5H glass coverslip bottom (ibidi, 81158) containing aCSF gassed with 95% $O_2$ and 5% $CO_2$ at room temperature. The microscopy dish was continuously perfused with the gassed aCSF via a peristaltic perfusion system PPS5 (Multi Channel Systems MCS GmbH) connected to a perfusion set (PECON, 500–800 173). Multichannel 3D time series were collected via a spinning disk confocal microscope (Nikon CSU-W1 SoRa), equipped with a 40×, 1.25 NA silicon objective. Dual color acquisitions were performed, with YFP and RFP excited using the laser wavelengths of 488 nm and 561 nm, respectively. The emitted light was collected via emission filters of 545/40 nm and 600/52 nm, respectively. The image dimensions in $xy$ were 1024 × 1024, and the voxel size was 0.162 × 0.162 × 0.9 μm. Each $z$-stack was recorded for 10–15 min with a 1-min interval between acquisitions.

## Immunohistochemistry

The mice were deeply anesthetized by intraperitoneal injection of xylazine (20 mg/kg body weight) and ketamine (200 mg/kg body weight) and transcardially perfused with physiological saline followed by 4% PFA in 0.1 M phosphate buffer, pH 7.4. The brains were dissected out of the skulls, placed into the same PFA buffer for 48 h at 4 °C, and then transferred to 30% sucrose in 0.1 M phosphate buffer, pH 7.4, until they sank. The brains were sliced into 40-μm-thick coronal sections using a sliding microtome (Leica), and the slices were stored at −20 °C in cryoprotectant solution (25% ethylene glycol, 25% glycerol, and 0.05 M phosphate buffer). The slices were incubated with the primary antibodies overnight at 4 °C and the secondary antibodies for 2 h (or 4 h for abberior STAR RED) at room temperature. The following primary antibodies were applied: anti-Iba-1 (rabbit, 1:500, Fujifilm Wako, 019-19741), anti-Ki-67 (rat, 1:500, eBioscience™, 14-5698-82), anti-P2RY12 (rabbit, 1:250, AnaSpec, AS-55043A). The following secondary antibodies were applied: donkey anti-rabbit IgG, Alexa Fluor™ 647 (1:400, Invitrogen, A31573), STAR RED (goat anti-rabbit IgG, 1:200, abberior, STRED-1002-500UG). For the morphological analysis, the slices were treated with the quenching buffer TrueBlack® (Biotium, #23007) with 2× solution following the manufacturer's protocol. Hoechst 33342 (1 μg/ml, Thermo Fisher Scientific, #H1399) was added for 10 min before mounting with mounting medium (Shandon™ Immu-Mount™, Epredia, #9990402). For super-resolution microscopy, a special mounting medium was applied (abberior MOUNT, SOLID ANTIFADE, abberior, MM-2013-2X15ML) followed by 1.5H glass coverslips.

## Confocal microscopy

For cell counting and Monte Carlo simulation analysis, the brain slices were imaged with the confocal laser scanning microscope Leica TCS SP5 with a 20×, 0.7 NA oil objective to capture 1024 × 1024 pixels images in $xy$. The voxel size was 0.638 × 0.638 × 2 μm. Six imaging channels were acquired for Hoechst, CFP, GFP, YFP, RFP, and Iba-1-AF-647 using laser wavelengths of 405 nm, 458 nm, 488 nm, 514 nm, 561 nm, and 633 nm, respectively. The corresponding emission signals were collected at 425–466 nm, 464–496 nm, 504–524 nm, 527–561 nm, 596–638 nm, and 655–755 nm, respectively. Three consecutive coronal slices from the center of the lesion (Bregma levels +0.145 to +0.545) were imaged with four images per hemisphere covering the lateral striatum area in the ischemic and the contralateral

sides. Microglial activation was confirmed based on the Iba-1 signal, independent of the Confetti markers.

For the Ki-67 analysis, the same excitation and emission settings were used but with a 40×, 1.3 NA oil objective. $z$-stacks of 1024 × 1024 pixels in $xy$ were acquired with a voxel size of 0.360 × 0.360 × 2 µm.

For visualizing microglial cell-cell interactions, the same setup and settings were used for the Hoechst, YFP, RFP, and Iba-1-AF-647 channels. Images were acquired from the ischemic region using a 63×, 1.4 NA oil objective and dimensions of 2048 × 2048 pixels in $xy$. The voxel size was 0.022 × 0.022 × 0.5 µm.

For the morphological analysis of microglia, the confocal laser scanning microscope LSM 700 (Zeiss) was used with a 40×, 1.3 NA oil objective to acquire 1024 × 1024 pixels images in $xy$. The voxel size was 0.156 × 0.156 × 1 µm. Images were acquired for the Hoechst and RFP channels. The GFP channel served as a reference to detect auto-fluorescence at the 8-week time point. Five images per hemisphere were acquired from five animals per time point (2 days, 1 week, and 8 weeks), resulting in 150 images in total.

### Stimulated emission depletion (STED) microscopy
Images were acquired with a Facility Line STED microscope from abberior Instruments equipped with a 60×, 1.42 NA oil objective. 405 nm, 488 nm, 561 nm, and 640 nm laser lines were used to excite Hoechst, YFP, RFP, and abberior STAR RED, respectively. A pulsed 775 nm STED laser was used for the depletion in the STAR RED channel. Overview $z$-stacks of 50 × 50 µm in $xy$ were first acquired with conventional confocal settings to fully visualize the interacting cells (voxel size 0.080 × 0.080 × 1 µm). This was followed by acquiring a $z$-stack of 10 × 10 µm in $xy$ with laser depletion of the STAR RED channel and a voxel size of 0.020 × 0.020 × 0.150 µm.

### Image processing and analysis
Cells' locations were extracted in a semi-automated manner using the Spots model in Imaris software v9.7. The automated detection was revised for all images, and errors in labeling were corrected manually. Imaris software was also used for surface rendering to visualize microglial cell-cell interactions and to create animated movies. For Ki-67 analysis, the surface rendering function was used to create a mask of the Confetti⁺ microglia. Cells were then classified based on the mean fluorescent intensity in the Ki-67 channel. The classification was manually corrected for false automated labeling.

For the morphological analysis of microglia, we trained a convolutional neural network, U-Net (2D), for image segmentation. We generated ground truth binary masks of nine $z$-stacks from the four investigated conditions using the LabKit plugin[64] in Fiji[65]. These ground truth masks were used to create datasets for training, validation, and testing. We split the $z$-stack masks and the corresponding raw data into patches of 256 × 256 pixels and used these patches as input for the U-Net model. 10% of the data was assigned as a testing dataset, and the remaining 90% was used to train the model with a 10% validation split. The model was trained for 100 epochs with a patch size of 25. After training and testing the model, it was implemented to segment the raw images of the whole dataset (150 $z$-stacks). Edge artifacts between the patches were eliminated by applying smooth blending of patches with 50% overlap. For the nuclear staining channel, we applied a series of steps to enhance the signal, followed by classical automated thresholding (Li), and morphological and size filters to isolate the nuclei within the microglia cells. This process was revised to manually exclude any objects that were falsely labeled as microglial nuclei. In addition, cells that largely cross the boundary of the image were excluded. The microglial nuclei were used as seeds for the 3D watershed from 3D Suite[66] to separate touching microglia. The 3D morphological features of each microglial cell were extracted by using the 3D region analysis from MorphoLibJ[67]. In addition, a 2D skeleton analysis was performed on maximum intensity projections using the

skeletonizing and 2D skeleton analysis from Fiji[65]. Highly correlated morphological parameters were excluded, and a selection of the five most informative morphological features was used to generate UMAP plots and perform statistical analyses.

### Computational analysis
The Monte Carlo simulation was performed in a similar approach to the established method in the literature[9,22]. The cell labels of all Confetti⁺ cells from one mouse were collected in one sampling vector, followed by bootstrapping. For each image, 1000 simulations were created by randomizing the cells' labels by sampling with replacement from the sampling vector of the corresponding animal. The locations of cells remained fixed. For each Confetti⁺ cell in the dataset, we placed concentric rings with radii ranging from 20 to 300 µm in 10 µm steps and a fixed width $w$ of 20 µm, e.g.,

$$r_i \in [20, 300]\,\mu m, r_i + 10\,\mu m = r_{i+1} \tag{1}$$

For each cell, the density of cells with the same color was calculated for all rings for the real recorded data and the 1000 corresponding Monte Carlo simulations. The simulated data for each condition were pooled, and the 2nd and 98th percentiles were calculated to yield the confidence interval. The cluster density for a specific cell and ring was calculated according to

$$d(c_n^j, r_i) = \frac{\left| \left\{ c_q^j \mid r_i - \frac{1}{2}w < \|c_n^j - c_q^j\|_2 \le r_i + \frac{1}{2}w, l_n^i = l_q^j, n \neq q \right\} \right|}{V(r_i, w)} \tag{2}$$

Where $j$ indicates the image. $n, q$ indicate the indices of the cells, and $l$ indicates the Confetti label. The ring volume is defined as:

$$V(r_i, w) = s_j \cdot 2r_i w \pi \tag{3}$$

Where $s_j$ is the height of the $z$-stack of image $j$. This equation stands for the entire ring, and it was adjusted in the case of partial coverage by the image.

Respectively, the densities were then averaged from all cells across images belonging to the same mouse, and then across mice belonging to the same time point and condition:

$$D(r_i) = \frac{1}{M} \sum_{m=1}^{M} \left( \frac{1}{J_m} \sum_{j=1}^{J_m} d(c_n^j, r_i) \right) \tag{4}$$

Where $M$ represents the number of mice and $J_m$ represents the number of images for mouse $m$. The densities were derived accordingly for the baseline recorded data and the Monte Carlo simulated data. Figure 2b, c demonstrates the recorded data against the Monte Carlo simulation data. When the recorded data lie within the range of the simulated data, it indicates a random distribution of the labeled cells in the tissue. Accordingly, deviation of the recorded data beyond the upper bound (98th percentile) of the simulated data indicates clonality.

Cluster analysis was performed using the DBSCAN algorithm (density-based spatial clustering of applications with noise). DBSCAN is an unsupervised clustering method that detects clusters by organizing data points according to their density, categorizing dense regions into clusters, and labeling outliers as noise. It runs based on the two assumptions of the minimum number of events per cluster and a radius of neighborhood. The radius of neighborhood was set to ($\varepsilon = 50\,\mu m$), and the minimum number of cells per clone was set to 2 cells. The average number of clones per brain hemisphere was calculated for each mouse. In addition, we estimated the average clone size of Confetti⁺ microglia by averaging the size of all clones from one

condition (stroke vs. contralateral) and a specific time point. For Ki-67 analysis, applying the DBSCAN algorithm was followed by calculating the proliferation index for each clone, which was defined as the number of Ki-67$^+$ cells within the clone divided by the clone size. The Monte Carlo simulation analysis and the cluster analysis were performed using MATLAB R2022a.

## Statistics and reproducibility

Data are presented as means ± s.e.m. or as medians ± IQR for parametric and nonparametric statistical tests, respectively, unless otherwise specified. Groups were compared using two-way ANOVA, followed by Šidák's or Dunnett's tests for multiple comparisons, or by the Kruskal-Wallis test followed by Dunn's test for multiple comparisons. Statistical analysis was done using GraphPad Prism 9 (GraphPad) or in Python using Pingouin[68] and scikit-posthocs[69] libraries. Differences were considered statistically significant with $P < 0.05$. Representative images presented in the manuscript are based on at least three independent replicates. Final figures were produced using Inkscape software v1.3 and Adobe Illustrator 2025.

## Reporting summary

Further information on research design is available in the Nature Portfolio Reporting Summary linked to this article.

## Data availability

The authors declare that the data supporting the findings of this study are provided within the manuscript and the supplementary materials. For running the code and reproducing analyses, an additional demo dataset was made publicly available here: https://doi.org/10.17605/OSF.IO/H9FJY[70]. Source data are provided with this paper.

## Code availability

All custom code needed to reproduce the analyses presented in this study was made available in this public GitHub repository: https://github.com/kikhiam/kikhia_et_al_2025[71].

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

## Acknowledgements

We would like to thank Melanie Kroh, Stefanie Balz, and Bettina Herrmann for their valuable technical assistance. We also acknowledge the

excellent support of the BIH Cytometry Core Facility, The Charité Core Facility for 7T Experimental MRIs, The NWFZ Light Microscopy Facility, The Advanced Medical BioImaging Core Facility of the Charité-Universitätsmedizin Berlin (AMBIO) and the AMBIO CBF satellite team, and the Scientific Computing of the IT Division at the Charité. We also thank Benjamin Judkewitz and Frank Heppner for the valuable discussions and their thoughtful input to this work. M.K. is supported by a PhD fellowship from the Einstein Center for Neurosciences Berlin. S.S. is funded by the Clinician Scientist Program of the Berlin Institute of Health (BIH). This work was supported by the Deutsche Forschungsgemeinschaft (Priority Program 2395/GE 2576/6-1 to K.G.; KR 2956/6-1 to G.K.; GE 2576/5-1 to K.G.; Germany´s Excellence Strategy—EXC-2049—390688087 to M.E.; Collaborative Research Center ReTune TRR 295- 424778381 to M.E.; Clinical Research Group KFO 5023 BeCAUSE-Y, project 2 EN343/16-1 to M.E.), the Bundesministerium für Bildung und Forschung (CSB to M.E., K.G., and G.K.), the German Center for Neurodegenerative Diseases (DZNE to M.E.), the German Center for Cardiovascular Research (DZHK to M.E. and K.G.), the German Center for Mental Health (DZPG to M.E.).

## Author contributions

M.K. conducted the imaging experiments, the flow cytometry experiments, and the computational analyses. S.S. conducted the patch-clamp experiments and analysis. M.L.H. conducted the MRI measurements. M.L. contributed to the methods and supervision of the computational analysis. T.L.T. and M.P. provided the mouse model and contributed to its implementation in this study. M.S. and H.K. contributed to the methods and supervision of the patch-clamp experiments. R.G. and K.G. conceptualized, designed, and supervised the study with contributions from G.K. and M.E. Additionally, R.G. and K.G. contributed to optimizing experimental protocols. The manuscript was written by M.K., S.S., R.G., and K.G. with contributions from all authors.

## Funding

## Competing interests

The authors declare no competing interests.

## Additional information

[1]Charité – Universitätsmedizin Berlin, corporate member of Freie Universität Berlin and Humboldt-Universität zu Berlin, Department of Neurology with Experimental Neurology, Charitéplatz 1, 10117 Berlin, Germany. [2]Charité – Universitätsmedizin Berlin, corporate member of Freie Universität Berlin and Humboldt-Universität zu Berlin, Center for Stroke Research Berlin (CSB), Charitéplatz 1, 10117 Berlin, Germany. [3]Charité – Universitätsmedizin Berlin, corporate member of Freie Universität Berlin and Humboldt-Universität zu Berlin, Einstein Center for Neurosciences Berlin, Charitéplatz 1, 10117 Berlin, Germany. [4]Berlin Institute of Health at Charité – Universitätsmedizin Berlin, Charitéplatz 1, 10117 Berlin, Germany. [5]DZHK (German Centre for Cardiovascular Research), partner site Berlin, Berlin, Germany. [6]Charité – Universitätsmedizin Berlin, corporate member of Freie Universität Berlin and Humboldt-Universität zu Berlin, Department of Neurosurgery, Predictive Modelling in Medicine Research Group, Augustenburger Platz 1, 13353 Berlin, Germany. [7]Max-Delbrück-Center for Molecular Medicine in the Helmholtz Association, Berlin, Germany. [8]Charité – Universitätsmedizin Berlin, corporate member of Freie Universität Berlin and Humboldt-Universität zu Berlin, Experimental Ophthalmology, Augustenburger Platz 1, 13353 Berlin, Germany. [9]Department of Biology, Boston University, Boston, MA, USA. [10]Department of Anatomy and Neurobiology, Boston University Chobanian & Avedisian School of Medicine, Boston, MA, USA. [11]Institute of Neuropathology, Faculty of Medicine, University of Freiburg, Freiburg, Germany. [12]Signalling Research Centres BIOSS and CIBSS, University of Freiburg, Freiburg, Germany. [13]Shenzhen Institutes of Advanced Technology, Shenzhen, China. [14]Charité – Universitätsmedizin Berlin, corporate member of Freie Universität Berlin and Humboldt-Universität zu Berlin, NeuroCure Cluster of Excellence, Charitéplatz 1, 10117 Berlin, Germany. [15]German Center for Neurodegenerative Diseases (DZNE), Partner Site Berlin, Berlin, Germany. [16]German Center for Mental Health (DZPG), Partner Site Berlin, Berlin, Germany. [17]Department of Adult Psychiatry and Psychotherapy, Psychiatric University Hospital Zurich, Zurich, Switzerland. [18]These authors contributed equally: Ria Göttert, Karen Gertz. ✉e-mail: karen.gertz@charite.de

