## [Transparent Peer Review file · Nature Communications]

Multicolor fate mapping of microglia reveals polyclonal proliferation, heterogeneity, and cell-cell interactions after ischemic stroke

Corresponding Author: Professor Karen Gertz

Version 0:

Reviewer comments:

Reviewer #1

(Remarks to the Author)

In this article, Kikhia et al study the clonal expansion and heterogeneity of microglia in ischemic stroke. The study aims to provide further granularity on the dynamics of microglia in stroke and, perhaps more importantly, to dissect whether a heterogenous response to stroke is in fact dictated by clonal origin. However, there are substantial issues related to the specific approach, which in my opinion are fundamentally precluding any of the key interpretations from being made (summarise in main point 1, below). Also, I felt this article is largely descriptive, lacking disease-relevant mechanistic insight.

Main points:

1. There is a fundamental issue with this study, which is the attribution of clonal identity based on the specific multicolour technique used here. The authors used the Microfetti mouse, which allows for 4 colour combinations to be stochastically expressed at time of labelling. This low number of colour combinations is not sufficient to unequivocally ascertain whether a 2 or more cells belong, or not, to the same clone, as cells could have acquired the same colour independently at time of labelling. This is further worsened by the fact that the microglia expressing the same colour are spatially intermixed, therefore making it even more difficult to say they belong to the same clone as they are not spatially related. These issues essentially preclude from making any attribution of functions shared by cells in the same clone, as the authors cannot say, unequivocally, that the cells in fact share the same clonal identity.
2. The doses of TAM used to induce recombination are pretty heroic. Please provide data on how many mice suffered adverse consequences, including mortality but also sickness behaviour. This is important to understand if the induction of the Microfetti model could condition the severity or recovery of the MCAo model, since no wild type mice used in this study.
3. What is the evidence about the 4 colours presenting the same frequency after recombination?
4. The data shown in fig3 is very interesting, but it's not clear what cells are being selected for patching and why. At some of those timepoints, it's well known some infiltrated monocytes will be present in the brain, how do you know you were patching microglia, specifically? If a method was in place, based on the microfetti tags, what was the sampling strategy (which cells from which colours)?
5. I felt this paper lacked clear mechanistic insight, as it's mostly the combination of two main approaches: 1) clonal marking linked to electrophysiological properties (Figs1/2/3/4), and 2) morphological analysis of microglia (figs5/6). These two elements are not clearly connected, and are mostly descriptive in nature. There is no evidence about the molecular nature of the changes observed in Figs5/6 and how they relate to the clonal analysis or the electrophysiological properties, nor there is a molecular explanation to how the clonal pattern is created and why that matters for disease.

Minor points:

1. The title is somehow fixated on the technical approach rather than the finding, can you please make it more specific?
2. The authors specifically mention that proliferation and apoptosis control numbers in homeostasis, but they need to cite the correct reference here Askew et al 2017, as the study cited in 7 is posterior.

(Remarks on code availability)

Reviewer #2

(Remarks to the Author)

In the paper by Kikhia et al., the authors explore the dynamics of microglial proliferation and cell-to-cell interactions in the model of moderate cerebral ischemia (30 mins MCAo) followed by reperfusion. By using computational analysis and multicolor fate mapping they established the temporal dynamics of clonal expansion of microglia after stroke starting from the acute post-ischemic response to the resolution phase. The changes in clonal dynamics have been then linked to electrophysiological and morphological properties of microglia. Although the concept of clonal expansion of microglia after brain injuries and neurodegeneration has been previously addressed, the presented data, owing to some technically original and innovative approaches, revealed some interesting insights into dynamics of microglia proliferation and development of associated electrophysiological and morphological phenotypes. Consequently, the most interesting observation is the time-dependent development of heterogeneity of microglial membrane properties between different clones during the late phase/resolution response after stroke. While, in general, the presented work is of interest, there are several points that need to be addressed:

1. Figure 1; The representative immunofluorescence images at different time points after stroke show several clones of cells expressing the different fluorescent markers. Based on the presented images it seems that some of the clones/colors are more prevalent than others. For instance blue and red vs. yellow. Please comment on this.
2. Back to the results presented in Fig. 1, this may suggest a distinct dynamics of proliferation between different clones. To clarify this issue, the authors may consider more in-depth analysis of microglia proliferation dynamics such as adding the ki-67 measurements and establish proliferation index for each clone.
3. Figures 4-5. Looking into heterogeneity of microglia starting after 8 weeks post-injury; is this event linked to inherent functional differences between distinct clones, and/or the microglial cells from different clones acquire different morphology that would over time necessarily lead to distinct membrane properties.
4. Figure 5. Here it is unclear whether microglial cells from all analyzed clones develop the same spectrum of morphological phenotypes. Also it is not clear to what extent the electrophysiological properties are determined by clonal origin or by developed morphology.... i.e. would two microglial cells sharing the same microenvironment and morphology, (but derived from different clones) exert the same functional/electrophysiological properties. Adding some algorithms may resolve this issue.
5. Few additional clarifications are needed regarding the experimental protocol used in this study. In the Methods Section the authors mention generation of the female Microfetti mice, however, the biological sex of experimental animals used in this study has not been clearly stated, and it is not clear whether experimental stroke has been performed on male and/or female mice.

(Remarks on code availability)

Reviewer #3

(Remarks to the Author)

In the present study, the authors use a powerful cellular fate mapping mouse model – the “Microfetti mice” - to analyze the responses of microglia to transient ischemic stroke. Combining immunohistochemical analyses with electrophysiological recordings from microglia, they provide a robust quantification of the time course of proliferative, morphological, and ion channel expression changes in microglia following stroke. The study also provides intriguing evidence that electrophysiological properties of microglia are driven by their clonal parentage and that microglia from distinct clones can potentially contact and interact with one another. Microglial responses to pathological insults are incredibly complex and understanding the mechanisms that underlie microglial properties and microglial cell-cell heterogeneity during pathology is extremely important for the field. Hence – the premise of this study is highly relevant. However, the study ultimately falls short on using these powerful approaches to their fullest and justifying the claim (repeated throughout the manuscript) that their study yields new insight into microglial stroke responses that can be used to guide new therapies.

Major concerns:

1) Although the Microfetti mouse enables elegant analysis of the lineage relationships of individual microglia, it's not clear that they have demonstrated clonal expansion in the way the title and written text claim. At present, the Monte Carlo simulation data (Fig. 2) is the only method employed to show whether clonal expansion is occurring, and there aren't any experiments that shed light on the functional relevance of microglial clonal expansion. In other words, it hasn't been shown unequivocally that microglial clonal expansion is happening and we don't learn anything about whether this cellular behavior is detrimental or beneficial in the context of stroke.

1a) Clonal expansion implies that daughter cells from recent cell divisions are more likely to proliferate than cells that have not recently undergone cell division. Given that the authors include computational modeling approaches in their study, it feels like they could have leveraged additional modeling approaches to investigate this. For example – if you know how much the overall microglial density increased by week 2 and you can estimate the number of microglial cell division events that would be needed to achieve that density increase – it seems that you should be able to model expected clone sizes if the probability of undergoing proliferation at any given time is equal for all microglia in the tissue (purely stochastic proliferation) or if microglia that recently experienced cell division are more likely to experience additional cell divisions (daughter cells have higher probability of continuing to proliferate – i.e. bias towards clonal expansion). This type of modeling can be used to predict whether the observed clone size of 6-8 cells during weeks 2 and 4 could only be achieved with a higher proliferative capacity of daughter cells or whether it is simply what would occur as a result of stochastic

microglial proliferation throughout the tissue.

1b) In the discussion, they describe how clonal expansion in the context of cancer involves a subset of cells gaining a competitive advantage over other cells and, hence, they expand to a greater degree. Throughout the manuscript, they seem to imply that this is what is occurring with microglial clones in the context of stroke, but additional experiments would be needed to show that expansion with a competitive advantage is occurring in microglial clones. For example, if we assume there is no bias in the original tamoxifen induced labeling of the microglia, then labeled cells with no competitive advantage for expansion should remain as individually labeled cells or clones of 2. Cells that possess some sort of competitive advantage should proliferate more dramatically, yielding larger clones. This would result in a bimodal distribution of clone sizes throughout the stroke tissue – tiny clones (1-2 cells = no competitive advantage) and large clones (8+ cells = some sort of competitive advantage). Can the authors show histograms of clone size from individual mice and reveal whether the distribution of clone size is normal or bimodal?

1c) In the discussion, the authors also describe clonal expansion of lymphocytes that have recognized their specific antigen, and the functional significance of this cellular behavior is clear. In contrast, the potential functional significance of any microglial clonal expansion in the context of stroke remains unclear. Is there any evidence that daughter cells that have undergone multiple, repetitive proliferative events are functionally different than the “older” cells in the tissue that have not experienced proliferation? This would be a super important finding for the field. Is the gene expression of cells that have experienced multiple proliferative events (members of clones containing 6-8+ cells) different than gene expression of individually labeled microglia (no proliferation) or 2 cells clones (minimal proliferation)? This could be investigated with in situ hybridization and a focus on key microglial inflammatory factors, trophic factors, phagocytosis-relevant genes or genes known to be involved in microglial-neuron or microglial-vascular interactions. One could also stain for CD68 or Clec7a or P2RY12 or similar and show whether all daughter cells in a large clone show signs of being functionally similar to one another and functionally distinct from surrounding microglia. With the current data – we have no insight into whether clonal expansion of microglia in the context of stroke is beneficial or detrimental.

2) Important methodological information is missing. 2a) The text of the results should describe the sex and age of the mice used for the study. Is it the case that only female mice were used in this study as the methods seem to indicate? What is the justification for that choice? What is the age of the mice when they are given tamoxifen? Stroke is predominantly a brain injury that impacts aged / aging individuals. While this reviewer appreciates that it would be impractical to carry out this study in aged mice, it is nonetheless critical to be transparent about whether these studies are being carried out in young adult vs. adult vs. middle aged mice. Older is clearly better for modeling the human stroke condition – so the authors should also include in the discussion a consideration of how the aging process may impact microglial responses to stroke. 2b) How are the authors defining regions of the dorsolateral striatum that are impacted by the stroke for acquiring images? The methods state that MRI analysis was used to confirm induction of stroke (this should also be described in the results section and ideally they should also include some sort of quantification of stroke severity to show how much animal-to-animal variation is present). However, once brain sections are prepared for confocal imaging and analysis, is a specific region of the dorsolateral striatum simply assumed to be impacted by stroke or do they use some type of staining in alternating brain sections to define the extent of stroke? It would be critical to know whether any cell-to-cell or clone-to-clone variation they are seeing is related to being fully within brain tissue impacted by stroke as opposed to being located at the boundaries of brain tissue impacted by stroke. For example, are the largest clones typically found within the center of tissue impacted by stroke while smaller clones are at the edges of tissue impacted by stroke?

3) Quantification needs to be included for the analysis of “interactions” between microglia in the same or different clones. What is the frequency of process to soma, process to process, soma to soma, and entangled interactions? Is there a relationship between these interactions and morphology of the cells involved? Is there a relationship between these interactions and proximity to stroke core versus edges of tissue impacted by stroke? Moreover, the data don’t actually show that microglia from the same or different clones establish “connections” with one another (lines 375-381). Showing clearly that cell-cell “contact” occurred requires super resolution imaging approaches (for example, STED or expansion microscopy) and ideally electron microscopy. Without these sorts of additional experiments, the language of manuscript should be modified to tone down this claim and just say that there are potential or putative contacts between microglia.

4) Few labs are equipped to carry out electrophysiological analyses of microglia, making these experiments a valuable addition to the literature. However, the field still doesn’t really know what the functional significance of microglial membrane currents is and the present study doesn’t really provide additional insight beyond describing observed membrane currents out to 8 weeks. Is there any relationship between size of the clone and the current profiles that are detected? This could suggest that there is a relationship between degree of recent proliferative events and membrane properties. Is membrane capacitance correlated with inward or outward currents, which might suggest relationships between membrane properties and morphological properties? For the microglial cells recorded in the contralateral hemisphere – were they recorded from all the different time points and then just pooled? Were there any subtle changes in number of cells that have the moderate inward conductance in the contralateral hemisphere across time points? There are almost certainly changes in the CSF milieu after injury and patterns of circuit activity beyond the stroke lesion cite will be altered. Hence, it would almost seem a bit surprising if there was NO change in the contralateral microglia after stroke. These types of data could provide insight into how much microglial electrophysiological properties are driven by immediate micro-environment versus more systemic status of the CNS.

Minor concerns:

1) For the t-SNE based analysis of morphology, did the authors examine which of the 28 extracted morphological features are highly correlated? Ideally they should show a correlation heatmap. Did they consider eliminating some highly correlated features before performing dimensionality reduction clustering? Otherwise, if they are including numerous morphological features that essentially capture the same thing about cell morphology, that morphological feature will be accorded undue

weight in the subsequent clustering.

2) The idea that microglia within one clone are “functionally homogeneous” (line 302 and lines 359, 360) seems like a stretch. The authors have shown that two recorded cells within one clone show similar membrane currents. However more information would be needed to say that they are functionally homogeneous. For example, are the cells that are within one clone also similar in terms of gene expression, cell process motility, phagocytic capacity? Without these sorts of additional experiments, the language of manuscript should be modified to tone down this claim.

3) For readers who are not in the stroke field, it would be helpful to provide a few sentences in the text of the results describing the 30min MCAO model and comparing it to other stroke models (relative severity at the tissue level – time course of any necrosis, blood brain barrier compromise and infiltration of peripheral cells; also relative severity at the behavioral level – does this model result in behavioral deficits comparable to those experienced by human stroke patients). This information will also hopefully allow the authors to justify why they chose this particular stroke model for their study. I may have missed it, but I don’t believe they gave any rationale for using this model.

4) Can the authors use the discussion to relate the findings from their various types of analysis? For example, how does the extent and time course of recovery of microglial morphological complexity after stroke compare to extent and time course of recovery of microglial electrophysiological properties after stroke? Are they similar? Is one faster? Are there relationships between typical morphological characteristics of cells and the size of the clone they are from?

5) Fig 1 – something looks very different about the 8W and 12W images – different scale? Is there more background fluorescence? Is there cellular debris?

6) Line 282 is likely a typo – probably should be “wrap” rather than “warp” around each other

7) Figure 4c – need to put “cell 1” and “cell 2” above the two examples of responses to hyperpolarization / depolarization to aid in reader understanding of the figure

(Remarks on code availability)

I do not have expertise in coding and hope that other reviewers can comment on this aspect of the study.

Version 1:

Reviewer comments:

Reviewer #1

(Remarks to the Author)

The authors have made an effort to justify their approach on the basis of the highlighted caveats on my initial review. However, the revision falls short by only providing justifications without resolving the actual issues. For example:

-Points 1 and 3: issues with microfetti mice. The authors provide further justifications but this does not modify the fact that their system is not capable of distinguishing if two cells that share colour belong to the same clone or just happened to be recombined to express the same colour (1/4 chance). This is particularly poignant for RFP cells (linked to point 3), as 40% of the recombined cells express RFP hence the likelihood of two cells expressing RFP out of chance is really high.

In short, the authors cannot provide a conclusive answer to this question, based on the used method, so all the downstream analysis is heavily compounded by this issue. The fact that the system has been used in other studies does not remove this caveat. If any, it adds the same issue to those published studies.

-Point 5: despite the explanations, and the new data provided, the morphological, clonal, and electrophysiological data is mostly descriptive and correlational. This study lacks mechanistic evidence connecting these multiple aspects, and most importantly lacks novelty in how it's advancing our understanding of the pathophysiology of stroke.

To sum up, I'm afraid I still believe this study does not merit publication, both on the grounds of technical flaws and lack of novelty.

(Remarks on code availability)

N/A

Reviewer #2

(Remarks to the Author)

The authors response to the raised critique is rather extensive and covered well the concerns raised by reviewers. At this time I do not have any additional concerns and/or questions.

(Remarks on code availability)

Reviewer #3

(Remarks to the Author)

In this revised manuscript, Kikhia and colleagues have added additional experiments and analyses that have greatly strengthened the study. In general, they have responded to most of the concerns raised by me and other reviewers. However, there is one key point that has not been fully resolved and, in my mind, is an important issue for the field. This is regarding the use of the term “clonal expansion.”

The classical, textbook definition of clonal expansion grew out of the cancer and immunology fields and involves thinking of cellular population dynamics as similar to Darwinian population dynamics of whole organisms. In the cancer field, this dates back to 1976, with Peter Nowell’s landmark perspective (PMID: 959840) arguing that individual somatic cells incur mutations that result in malignant transformation. They then proliferate and begin to outnumber healthy neighboring cells. Because all the malignant cells are daughters of an initial parent that incurred mutations, this represents a “clonal expansion” by which numerous daughter cells with some key defining attributes come to dominate the local cellular population. Publications in the cancer field continue to use the term “clonal expansion” to refer to a specific group of cells with unique, identifiable attributes that have become overrepresented in the local cell population through proliferation and a Darwinian, natural selection like process (for example PMID: 17109012 and PMID: 28912577).

The same principles are evident in Immunology, where the classic, textbook example of clonal expansion is the behavior of lymphocytes. In 1957 and 1959, David Talmage and Sir Frank Macfarlane Burnet proposed that lymphocyte function and behavior were driven by a “selection process” (PMID 13425332, and “The Clonal Selection Theory of Acquired Immunity, Vanderbilt Univ. Press, 1959). The idea was that lymphocytes stochastically express numerous diverse antigen receptors and that when a lymphocyte encounters a ligand for its particular antigen receptor, this drives a massive proliferation and differentiation of those specific cells. Again – through massive proliferation beginning with one or few parent cells, these daughter lymphocytes with a unique set of attributes come to (transiently) dominate the overall population of lymphocytes. This is the essence of clonal expansion and is the manner in which the immunology field (at least the vast majority of the field) continues to use this term (PMID 32424244, PMID 38506411).

Other fields have used the term “clonal expansion” with the same underlying meaning that a specific subset of cells with unique attributes come to dominate the overall cell population through proliferation from very few parent cells. This includes the cardiovascular field and formation of atherosclerotic plaques (PMID: 38362345) where one of the studies the authors cite (Chappell et al 2016, PMID: 27682618) uses Microfetti to show that plaques are dominated by progeny of one or two clones. They state “In contrast to the mosaic stochastic labeling observed in the vascular wall, VSMC-derived cells within plaques were found in large monochromatic regions with little intermixing between colors.” They conclude that “extensive proliferation of a low proportion of highly plastic VSMCs results in the observed VSMC accumulation after injury and in atherosclerotic plaques.”

Hence, the essential elements of the process of clonal expansion are 1) that cells proliferate to give rise to daughter cells and 2) that a specific group of cells with some unique attribute (activated antigen receptor, malignant transformation, unusually high proliferative potential) outcompetes the other local cells and becomes a huge clone that is over-represented within the local cellular environment/cellular population. In other words, to apply the term “clonal expansion,” it’s not enough to show that individual clones increase in size/cell number. You need to show that specific clones expand RELATIVE TO OTHER CLONES and/or relative to the local cell population.

The present study and other studies that have used microfetti mice to study microglia and macrophages (Tay et. al, 2017, Zhan et. al 2019, Jordao 2019, Masuda 2022) have used this tool to show convincingly that microglia/macrophages proliferate in various contexts. This proliferation results in clones with roughly 2-25 same color daughter cells. This allows them to move beyond what is possible with BrdU labeling or single color lineage tracing to show how many cell divisions typically arise from an individual parent, to explore whether neighboring clones have distinct properties or disparate capacity to support health and function of nearby synapses/neurons etc, and to see if cells from neighboring clones intermix or interact with one another.

Microfetti can also be used to see if any clones come to be overrepresented or dominate the local environment. However, none of the studies listed above, nor the present study have not shown that any microglial clones appear to have some sort of competitive advantage over other clones or that specific types of microglial clones become over-represented and dominate the local cellular environment. Showing that clones within or near an area of pathology proliferate more than clones distant from pathology is not the same as showing that – within the pathology zone - specific clones have elevated proliferative capacity or a competitive advantage relative to other clones that are also in the pathology zone. If you haven’t shown this, then you haven’t met the criteria for claiming that clonal expansion is occurring in the context of that pathology.

In their rebuttal, the authors point to use of Monte Carlo simulation as a statistical approach to show that clonal expansion is occurring. This is also inaccurate. In the words of Tay et al, Monte Carlo simulation allowed them to “infer whether same-color microglia were related as progeny of proliferation events or were located in the same neighborhood simply due to chance.” In the words of Cabeza – Cabrerizo 2019, Monte Carlo simulation was used “to determine the probability that the observed clusters could have arisen by chance.” In other words, Monte Carlo simulation can confirm that your observed cell distribution of cells of different colors occurred as a result of proliferation of parent cells and location of daughter cells near to the parent, rather than simply due to chance. But this does not prove that a process of “clonal expansion” – as it is classically defined – is occurring.

I strongly believe that the current manuscript makes very valuable additions to our understanding of the dynamics of microglial responses to MCAo. They show that daughter cells that are part of the same clone share key properties (e.g. potassium channel expression) that can differ from those of cells in a neighboring clone. They reveal key nuanced details about the dynamics of microglial proliferation, about some of the mechanisms that can underlie cell to cell heterogeneity during responses to MCAo, and the dynamics of returning microglia to baseline numbers during the resolution phases of responses to pathological insults. Even if none of the microglial clones have a competitive advantage and are not undergoing classical “clonal expansion,” it will be very important to understand if clones with different properties (e.g. potassium channel expression) have a greater or lesser capacity to support local synapse/neuron health and tissue repair.

But I also strongly feel that the present and recent microglial studies are using the term “clonal expansion” in a way that may confuse the reader and imply that certain cellular population dynamics similar to those observed in immunology and cancer are occurring when they are not. In my opinion, the authors should not use this term to describe their data and should instead use some sort of term like “clonal dynamics” or “clonal heterogeneity.” Of course, it will be up to the editors to make a final decision on this matter. At the very minimum, there needs to be a clear discussion of the classical definitions of clonal expansion earlier in the paper (introduction) and very clear statements that they do not have evidence that specific microglial clones exhibit any competitive advantage relative to other clones during responses to MCAo, so the clonal microglial dynamics they are observing differ from what is observed in immunology and cancer.

I apologize if this long discussion seems pedantic, as that was not my intent. This is an excellent and important study that can further do a service to the field by being very careful and judicious in use of terminology.

(Remarks on code availability)

Rigorous checking of the code is not my area of expertise - hopefully other reviewers are able to help in this regard.

Version 2:

Reviewer comments:

Reviewer #3

(Remarks to the Author)

The authors have very thoughtfully considered the most appropriate terminology to accurately describe their results regarding microglial responses following MCAO. These revisions have addressed my remaining concerns.

(Remarks on code availability)

n/a

Response to referees:

We sincerely thank all reviewers for their time and effort and appreciate their detailed and constructive feedback on our work. Accordingly, we have done our best to address all comments raised by the reviewers within the timeframe of the revision process. Here, we provide a point-by-point response to all reviewers' comments. The changes in the updated manuscript are highlighted in red. Reviewers' comments are presented in *italic* format.

Reviewer #1:

In this article, Kikhia et al study the clonal expansion and heterogeneity of microglia in ischemic stroke. The study aims to provide further granularity on the dynamics of microglia in stroke and, perhaps more importantly, to dissect whether a heterogenous response to stroke is in fact dictated by clonal origin. However, there are substantial issues related to the specific approach, which in my opinion are fundamentally precluding any of the key interpretations from being made (summarise in main point 1, below). Also, I felt this article is largely descriptive, lacking disease-relevant mechanistic insight.

Response:

We thank the reviewer for their critical review and for pointing out the relevance and the limitations of our study. We fully acknowledge the limitations mentioned and we have done our best to address them in the revised manuscript. In addition, we meticulously rephrased our conclusions to match the strength of the provided evidence. We also agree with the reviewer that our study is largely descriptive. However, we strongly believe that such descriptive studies have a valuable place in literature. Several preclinical interventional studies aimed to target microglia as a potential treatment of stroke, leading however to conflicting results¹⁻³. The main challenge in therapeutically targeting microglia is to reduce harmful mechanisms while maintaining and enhancing protective mechanisms. To achieve this, a deeper understanding of microglial dynamic response to stroke is necessary. Recent single-cell sequencing studies, which are also descriptive in nature, have shown the complex transcriptional realm of microglia. However, little has been done to provide a detailed description of the complex behavior of microglia after stroke. In our current study, we used a plethora of advanced methods to highlight key elements of that response. We strongly believe that such work should precede interventional studies to establish the necessary knowledge for effective interventions. We emphasized this in the revised manuscript.

Main points:

1. There is a fundamental issue with this study, which is the attribution of clonal identity based on the specific multicolour technique used here. The authors used the Microfetti mouse, which allows for 4 colour combinations to be stochastically expressed at time of labelling. This low number of colour combinations is not sufficient to unequivocally ascertain whether a 2 or more cells belong, or not, to the same clone, as cells could have acquired the same colour independently at time of labelling. This is further worsened by the fact that the microglia expressing the same colour are spatially intermixed, therefore making it even more difficult to say they belong to the same clone as they are not spatially related.

These issues essentially preclude from making any attribution of functions shared by cells in the same clone, as the authors cannot say, unequivocally, that the cells in fact share the same clonal identity.

Response:

We agree with the reviewer that the four-color labeling strategy has limitations. We have addressed this issue in the discussion section of the original manuscript (currently lines 401-406). Despite this limitation, the Microfetti mouse offers several advantages making it a suitable model for our study among the available options. These advantages are:

1. Effective labeling of microglia after a single i.p. injection of tamoxifen.
2. The system provides specific, reproducible, stable, and dose-dependent expression of endogenous reporter proteins in the microglia of the adult mouse brain.
3. The system makes it possible to exclude contamination of microglia data by confetti⁺ infiltrating monocytes/macrophages from blood, which is very important for a stroke study.
4. The model has already been tested and validated in a neurodegenerative environment (facial nerve axotomy⁴)

To reduce the probability of spatial intermixing, we used a tamoxifen dosage that results in sparse labeling of microglia, i.e., only about 17% of Iba-1 positive cells in the contralateral striatum expressed fluorescent reporter proteins (See **Fig. 2a** and the figure below). In addition, under homeostatic conditions in the contralateral striatum, the average distance to the nearest neighbor cell of the same color is $137.5 \pm 7.6 \mu\text{m}$. Therefore, microglial cells in stroke tissue that have the same color and are less than $50 \mu\text{m}$ apart are very likely part of the same clone.

Additional Fig. 1 Sparse labeling of microglia with Confetti fluorescent reporter proteins in the non-ischemic contralateral striatum at different time points after MCAo.

We would like to emphasize that highly impactful studies from different groups have applied similar Confetti systems to study the clonality of different cell populations. Here is a non-extensive list of such studies:

- Microglia: Tay *et al.* (2017)⁴, Zhan *et al.* (2019)⁵, Jordão *et al.* (2019)⁶
- CNS-associated macrophages: Masuda *et al.* (2022)⁷
- Conventional dendritic cells: Cabeza-Cabrerizo *et al.* (2019)⁸
- B lymphocytes: Tas *et al.* (2016)⁹
- Intestinal stem cells: Snippert *et al.* (2010)¹⁰, Schepers *et al.* (2012)¹¹
- Enteric nervous system progenitors: Lasrado *et al.* (2017)¹²
- Vascular smooth muscles: Chappell *et al.* (2016)¹³

Similar to those studies, we followed the well-established method of comparing recorded data with stochastic simulated data using Monte Carlo simulations. This analysis proves the clonal expansion of microglia and is not compromised by the four-color resolution of the system.

Other multicolor fate mapping approaches can offer more color combinations based on RGB marking^{14,15}. Such RGB marking can facilitate deeper clonal analysis, but the currently available microglia-specific systems have limitations, too. One potential disadvantage is intracerebral viral vector delivery, which may result in alterations to the extracellular matrix at the injection site and needle-track tissue damage with the potential disruption of the blood-brain barrier. These damages may induce the priming of immune cells, which could impact the heterogeneity of microglia phenotypes following MCAo. Consequently, it would have been necessary to include further control groups in the study and increase sample sizes to accommodate this potential outcome. Furthermore, the viral delivery may exhibit a gradient in the brain, resulting in an uneven distribution of reporter expression. Finally, such approaches may result in off-target labeling with a microglia specificity of $78.36\% \pm 12.06\%$ in the adult mouse brain¹⁵. Recently, a less invasive, microglia-targeted AAV technology has been reported¹⁶⁻¹⁸. In further studies, those approaches in combination with RGB constructs might be an option to label and trace microglia after brain ischemia. Another option would be to create transgenic mice with microglia-specific combinatorial fluorescent protein expression using the Brainbow-1.0 construct, resulting in a non-invasive and more specific transgenic RGB system¹⁹. In sum, future studies would benefit from microglia-specific, non-invasive, RGB marking systems. We expanded the discussion of this point in lines 401-406.

2. The doses of TAM used to induce recombination are pretty heroic. Please provide data on how many mice suffered adverse consequences, including mortality but also sickness behaviour. This is important to understand if the induction of the Microfetti model could condition the severity or recovery of the MCAo model, since no wild type mice used in this study.

Response:

We thank the reviewer for the comment. The dosage we used in this study is within the range used by other studies in the field. Faust and colleagues applied a dosage of 100 mg/kg for four consecutive days in their comparative analysis of microglial inducible Cre lines²⁰. Huang *et al.* (2018) applied 150 mg/kg of body weight for a few consecutive days²¹. Zhan *et al.* (2019) applied 2 mg per day for 4 consecutive days⁵. We would like to clarify that a single dose of tamoxifen was administered intraperitoneally at a dose of 250 mg/kg body weight. For instance, a mouse weighing 20 g received a single injection of 5 mg of tamoxifen, which was the same dosage used previously in adult animals carrying the confetti construct¹⁰. It should be noted that this corresponds to about half of the dose used in Tay *et al.* (2017) (single injection of 10 mg s.c.)⁴. We added further clarification on the tamoxifen dosage in lines 431-433.

Induction and consequent expression of confetti fluorescent proteins depend on many factors, including the organ of interest, the promoter activity of the chosen Cre-ERT2 driver line as well as the tamoxifen formulation and tamoxifen administration route. Therefore, tamoxifen dosage has to be evaluated for every mouse model and biological question independently. With the chosen dosage we reach a recombination efficiency of 17% of microglia. We believe that a single injection paradigm is the most appropriate for studies related to microglial clonal expansion after MCAo using the confetti construct. As it was shown, a second dosage of tamoxifen can induce a flip in the remaining floxed structure and result in flipping from one color to another¹¹.

Please note that a six-week waiting period was implemented between the tamoxifen injection and MCAo surgery. This waiting period was implemented to exclude contamination of microglia data by blood-derived confetti⁺ monocytes/macrophages invading the stroke lesion (**Supplementary Fig. 1b**). Mice that underwent MCAo showed no sickness behavior after tamoxifen administration, as the weight gain of animals indicates (see figure below). Furthermore, the location and size of infarct lesions in Microfetti mice were comparable to lesions of untreated WT mice in other studies of our lab. We have added an example MRI image and T2-MRI lesion volumes in **Supplementary Fig. 1c**. Therefore, we do not believe that the induction of the Microfetti model could affect the severity or recovery of the MCAo model.

Additional Fig. 2 Weight gain of Microfetti mice in the waiting period between tamoxifen injection and MCAo surgery.

3. What is the evidence about the 4 colours presenting the same frequency after recombination?

Response:

We thank the reviewer for this important question. In the Microfetti mouse, the expression of the Confetti markers is a stochastic process that relies on the location of the *Cre* recombinase activity and the direction of the final stable construct as shown in (**Supplementary Fig. 1a**). However, as pointed out by reviewer 2, we have observed in our flow cytometry and imaging data that the frequency of the four confetti colors is unequal. The figure below shows the percentage of the four colors in Confetti⁺ microglia for all animals used to generate the imaging data (**Fig. 1** and **Fig. 2**). This data shows that RFP (39.81±1.37%) and YFP (31.65±0.98%) are more frequent than CFP (19.72±1.27%), and GFP (8.81±0.95%). Such a pattern was also observed in previous studies using the Microfetti mouse^{4,5}. The underlying mechanism of this pattern is unknown. Nevertheless, we would like to emphasize at this

point that an even recombination of the four Confetti markers was never an assumption for our analyses. Instead, the specific frequencies observed for each color in each mouse were utilized to perform the Monte Carlo simulation. Therefore, an equal recombination frequency is not a necessary condition for conducting the analyses.

Additional Fig. 3 The percentage of the four colors among all Confetti⁺ cells in the animals included in (**Fig. 1** and **Fig. 2**).

4. *The data shown in fig3 is very interesting, but it's not clear what cells are being selected for patching and why. At some of those timepoints, it's well known some infiltrated monocytes will be present in the brain, how do you know you were patching microglia, specifically? If a method was in place, based on the confetti tags, what was the sampling strategy (which cells from which colours)?*

Response:

We thank the reviewer for showing high interest in our data. As outlined in the methods part of the original manuscript (currently lines 493-494): “YFP was excited at 480 nm, emitting light at 521-561 nm; RFP was excited at 550 nm, emitting light at 580-630 nm”), we used the YFP and RFP signal to identify cells for patch-clamping. We have chosen these two Confetti markers because their cytoplasmic distribution provides good visualization of the microglial cell and its borders to facilitate efficient patching. To clarify our sampling strategy for patch-clamp recordings and how we defined clones and neighbor cells in the ischemic tissue, we added the following sentence to the methods part in lines 488-493: “YFP and RFP signals were utilized to identify microglia because the cytoplasmic distribution of the two reporter proteins allows adequate visualization of the cell and its borders, facilitating efficient patching. Clones were identified as microglial cells expressing one color (YFP or RFP) and having a distance of approximately 50 μm from the nearest neighbor. Neighboring cells of different clones were identified as cells expressing different colors and being less than 50 μm apart.”

It is well established that blood-derived monocytes/macrophages infiltrate ischemic brain tissue. This phenomenon has been previously examined in our laboratory in a separate study²². As described above, it should be noted that a six-week waiting period was implemented between the tamoxifen injection and MCAo surgery. Monocytes in the blood also express the confetti colors after the tamoxifen injection. However, since monocytes have a short half-life of only a few days, there are no confetti⁺ monocytes left in the blood at the time of MCAo that could invade the stroke tissue. We verified this by analyzing blood cells using flow cytometry and were able to show the absence of Confetti⁺ monocytes five weeks after the tamoxifen injection (**Supplementary Fig. 1b**). This significant issue was previously addressed in the initial submission (currently lines 93-96).

5. *I felt this paper lacked clear mechanistic insight, as it's mostly the combination of two main approaches: 1) clonal marking linked to electrophysiological properties (Figs1/2/3/4), and 2) morphological analysis of microglia (figs5/6). These two elements are not clearly connected, and are mostly descriptive in nature. There is no evidence about the molecular nature of the changes observed in Figs5/6 and how they relate to the clonal analysis or the electrophysiological properties, nor there is a molecular explanation to how the clonal pattern is created and why that matters for disease.*

Response:

We thank the reviewer for the critical comment. Indeed, discovering the underlying patterns of correlations between the morphology and the functions of microglia is one of the main current goals of the field²³. However, the current available evidence suggests that this relationship is complex and it is recommended to not simply infer function based on morphology alone²³. It was indeed a challenge for us to combine accurate morphological and functional readouts on a single-cell level. The patch-clamp setup in our lab is equipped with a standard epifluorescence microscope that allows us to take single z-plane images (as shown in **Fig. 4**) but does not allow for z-stack acquisition for accurate 3D reconstruction. On the other hand, we have access to multiple confocal systems that allow high-resolution 3D reconstruction, but none of them are equipped with electrophysiological setups. Therefore, the two analyses had to be conducted separately. Despite this limitation and as mentioned by reviewer 2, only a few laboratories are able to perform electrophysiological studies on microglia,

which makes our study a valuable addition, especially for stroke research. In principle, we have identified important parallels between the two analyses, although being conducted independently. For example, the changes in capacitance depicted in **Fig. 3c** map very well to the changes in “surface area/volume” in **Fig. 5b**. Furthermore, the presence of microglial cells with small currents after 8 weeks similar to the contralateral side (**Fig. 3e**) also matches our morphological data where highly ramified cells reappear after 8 weeks but not after 2 days or 1 week (**Fig. 5c**).

In the revised manuscript, we added a new electrophysiological experiment (**Fig. 4**), where we compared the electrophysiological properties of neighbor cells from different clones and found them to be less similar than cells belonging to the same clone (neighbors with the same color). This experiment emphasizes the relevance of the clonal expansion of microglia and provides an additional important link between the electrophysiology experiments and the clonal analysis. This analysis is novel and provides valuable insight for future studies that can look into its molecular mechanism. We highlighted this in lines 242-255.

Motivated also by the suggestion of reviewer 3, we revised the discussion section to link the different analyses (see lines 350-368).

For a detailed response on the relevance of this study to disease, please see to our response to reviewer 3, major concern 1.

Minor points:

1. *The title is somehow fixated on the technical approach rather than the finding, can you please make it more specific?*

Response:

The current title was a result of a lengthy discussion among the authors. We found this title to be the best summary we could come up with for the current manuscript.

2. *The authors specifically mention that proliferation and apoptosis control numbers in homeostasis, but they need to cite the correct reference here Askew et al 2017, as the study cited in 7 is posterior.*

Response:

We thank the reviewer for reminding us of this important reference. **We cited it accordingly in line 59.**

Reviewer 2:

In the paper by Kikhia et al., the authors explore the dynamics of microglial proliferation and cell-to-cell interactions in the model of moderate cerebral ischemia (30 mins MCAo) followed by reperfusion. By using computational analysis and multicolor fate mapping they established the temporal dynamics of clonal expansion of microglia after stroke starting from the acute post-ischemic response to the resolution phase. The changes in clonal dynamics have been then linked to electrophysiological and morphological properties of microglia. Although the concept of clonal expansion of microglia after brain injuries and neurodegeneration has been previously addressed, the presented data, owing to some technically original and innovative approaches, revealed some interesting insights into dynamics of microglia proliferation and development of associated electrophysiological and morphological phenotypes. Consequently, the most interesting observation is the time- dependent development of heterogeneity of microglial membrane properties between different clones during the late phase/resolution response after stroke. While, in general, the presented work is of interest, there are several points that need to be addressed:

Response:

We sincerely thank the reviewer for appreciating the value of our work and for the comprehensive and constructive feedback.

1. Figure 1; The representative immunofluorescence images at different time points after stroke show several clones of cells expressing the different fluorescent markers. Based on the presented images it seems that some of the clones/colors are more prevalent than others. For instance blue and red vs. yellow. Please comment on this.

Response:

We thank the reviewer for this comment. We addressed this finding in our response to reviewer 1. Please see our text above with the corresponding figure in our response to “main point 3/reviewer 1”.

2. Back to the results presented in Fig. 1, this may suggest a distinct dynamic of proliferation between different clones. To clarify this issue, the authors may consider more in depths analysis of microglia proliferation dynamics such as adding the ki-67 measurements and establish proliferation index for each clone.

We thank the reviewer for this valuable suggestion. Indeed, the observation of different clone sizes could reflect distinct proliferation dynamics. To visualize the clone size dynamics over time more clearly, we have created another pie chart (see figure below). The majority of clones remain rather small, with up to five cells. However, clones with more than 20 cells were found, albeit in small numbers, at 4 weeks post-stroke.

Additional Fig. 4 Clone sizes observed in the ischemic dorsolateral striatum at different time points after MCAo.

To look deeper into this issue, we followed the suggestion of applying the Ki-67 measurement to establish a proliferation index for each clone. We added the results of this analysis in **Supplementary Fig. 2**. The data indicates that approximately 40% of Confetti⁺ microglia in the stroke region were Ki-67⁺ two days after MCAo. This percentage diminished to approximately 20% one week after MCAo and declined further to less than 10% after two weeks. When we analyzed the absolute numbers of Confetti⁺/Ki-67⁺ microglia, we found that the peak was reached one week after MCAo (**Supplementary Fig. 2b**). Consistent with the data presented in (**Fig. 1** and **Fig. 2**), these additional new data indicate that the majority of microglial proliferation following a stroke occurs within the first two weeks. However, some microglial cells continue to proliferate after two weeks.

A thorough examination of the proliferation index for individual clones (**Supplementary Fig. 2c**) revealed a multitude of intriguing dynamics. As expected after two days, we only observed singlets or small clones (2-5 cells) with highly variable proliferation indices of 0-100%. This simply indicates that at two days some cells have already started the proliferation process forming small clones and some haven't started this process yet. At one week, it becomes evident that the relationship between clone size and proliferation index is inversely proportional. Consequently, only small clones (2-5 cells) exhibited a proliferation index of 100%. The fact that many clones show multiple Ki-67⁺ cells indicates that a clone cannot be the result of a single mother cell dividing on its own, but rather that the respective daughter cells also possess the capacity to proliferate. A marked decline in the proliferation index of the clones was observed between the first and second week after MCAo. This results in the observation of larger clones at two weeks with fewer proliferating, Ki67⁺ microglial cells. Across all time points, a high degree of variability in the proliferation index was observed, thereby confirming a high degree of heterogeneity in the proliferation capacity of different clones. These results were added to the manuscript in the lines **172-177**.

3. Figures 4-5. Looking into heterogeneity of microglia starting after 8 weeks post-injury; is this event linked to inherent functional differences between distinct clones, and/or the microglial cells from different clones acquire different morphology that would over time necessarily lead to distinct membrane properties.

Response:

We thank the reviewer for raising this important question. As previously stated in response to reviewer 1 (main point 5), the current data does not allow for definitive conclusions regarding the correlation between specific current profiles and specific morphological profiles. This is due to the inability to ascertain the morphology of the patched cells with the same level of precision as the U-Net-based morphological analysis. Filling microglia cells with fluorescent dyes in acute slices during or after patch-clamp analysis does not yield high-resolution pictures. In the past, we have fixed slices after recording and tried to find the recorded cell and analyze its morphology. The success rate is usually low, in the range of one or two out of ten. When the patch-pipette is retracted, the cell is often damaged and leaks out.

Nevertheless, our data shows a significant increase in capacitance two days after stroke (**Fig. 3c**). This increase indicates a decrease in "surface area/volume" as seen in (**Fig. 5b**). We observed a correlation between this particular morphological change and the presence of the delayed outward rectifying current, which is described in (**Supplementary Fig. 3a**). Interestingly, the presence of another outward current at eight weeks (described in lines 236-241) did not correlate with an increase in capacitance. As outlined in lines 343-347 of the discussion, delayed outward rectifying currents, such as Kv1.3, have been associated with stroke pathology and proinflammatory microglial phenotypes. However, further research is needed to understand the correlation between specific current profiles and specific morphological properties of microglia *in vivo*.

4. Figure 5. Here it is unclear whether microglial cells from all analyzed clones develop the same spectrum of morphological phenotypes. Also it is not clear to what extent the electrophysiological properties are determined by clonal origin or by developed morphology.... i.e. would two microglial cells sharing the same microenvironment and morphology, (but derived from different clones) exert the same functional/electrophysiological properties. Adding some algorithms may resolve this issue.

Response:

We thank the reviewer for raising this important point. On this basis, we conducted additional experiments that further substantiated our hypothesis and strengthened the substantive message of

the project. To do so, we conducted a new patch-clamp experiment in which we patched neighboring cells from different clones, i.e. neighboring cells that were <50µm apart and had different Confetti markers. The results are presented in (Fig. 4 and Supplementary Fig. 4b). Interestingly, we observed neighboring cells from different clones are less similar in their electrophysiological profiles than cells belonging to the same clone. Some of the neighboring cells from different clones show very distinct electrophysiological profiles (Fig. 4c, neighbors A and C), while others show similarities (Fig. 4c, neighbors B and D). In contrast, cells belonging to the same clone exhibit nearly identical profiles (Fig. 4b and Supplementary Fig. 4a). These results suggest a higher degree of functional correlation within a single clone, thereby proposing an effect of clonality on microglial function. It is important to emphasize that these results do not negate the role of the microenvironment. It is well established that microglia interact with and respond to their microenvironment. However, our new data suggests that microglia can inherit phenotypical features from their mother cells, resulting in a higher degree of functional correlation within a single clone. This point is further elaborated in the discussion, specifically in lines 316-340. The result section on electrophysiological properties contains the new experiments and has been rewritten in lines 242-255.

5. Few additional clarifications are needed regarding the experimental protocol used in this study. In the Methods Section the authors mention generation of the female Microfetti mice, however, the biological sex of experimental animals used in this study has not been clearly stated, and it is not clear whether experimental stroke has been performed on male and/or female mice.

Response:

The manuscript has been revised in accordance with the provided feedback (line 424-426). This study utilized female Microfetti mice. For a more thorough exposition, please refer to the detailed responses provided to reviewer 3, specifically addressing “major point 2a”.

Reviewer 3:

In the present study, the authors use a powerful cellular fate mapping mouse model – the “Microfetti mice” - to analyze the responses of microglia to transient ischemic stroke. Combining immunohistochemical analyses with electrophysiological recordings from microglia, they provide a robust quantification of the time course of proliferative, morphological, and ion channel expression changes in microglia following stroke. The study also provides intriguing evidence that electrophysiological properties of microglia are driven by their clonal parentage and that microglia from distinct clones can potentially contact and interact with one another. Microglial responses to pathological insults are incredibly complex and understanding the mechanisms that underlie microglial properties and microglial cell-cell heterogeneity during pathology is extremely important for the field. Hence – the premise of this study is highly relevant. However, the study ultimately falls short on using these powerful approaches to their fullest and justifying the claim (repeated throughout the manuscript) that their study yields new insight into microglial stroke responses that can be used to guide new therapies.

Response:

We thank the reviewer for highlighting the potential relevance of our study to the field and for the detailed and constructive revision of our work. We hope that we have not overstated the relevance of our study in the original manuscript. Therefore, we have carefully revised our statements related to stroke therapy and linked them directly to the results of the current study (lines 53-54, 337-340, 416-418)

Major concerns:

1) Although the Confetti mouse enables elegant analysis of the lineage relationships of individual microglia, it's not clear that they have demonstrated clonal expansion in the way the title and written text claim. At present, the Monte Carlo simulation data (Fig. 2) is the only method employed to show whether clonal expansion is occurring, and there aren't any experiments that shed light on the functional relevance of microglial clonal expansion. In other words, it hasn't been shown unequivocally that microglial clonal expansion is happening and we don't learn anything about whether this cellular behavior is detrimental or beneficial in the context of stroke.

Response:

Although microglia clonal expansion is a relatively new concept, there is currently no doubt that microglia show this behavior in different contexts. The current study builds on previous literature that confirmed microglial clonal expansion in response to facial nerve axotomy⁴, experimental autoimmune encephalomyelitis⁶, replenishment after depletion⁵, and during development¹⁵. In the current study, we show that the number of Confetti⁺ microglia significantly increases in the stroke region one week after MCAo (**Fig. 1e**). This increase can already be seen visually through the formation of clusters of cells sharing the same color (**Fig. 1d**), which clearly reflects clonal expansion. To confirm this mathematically, we applied the well-established method of comparing the recorded data with stochastic simulated data via Monte Carlo simulations. When the recorded data shows higher densities than the simulated data, as observed in **Fig. 2c**, this already confirms clonal expansion as has been shown by other studies using Confetti mouse lines and Monte Carlo simulations^{4,6,8}. Using a second independent approach, we applied the clustering algorithm DBSCAN, which clusters the cells in our data based on their color and spatial distribution. This analysis also mathematically confirms the emergence of clones in the stroke region within the first week after MCAo (**Fig 2d, e**). As requested by the reviewer, we also provide histograms of clone sizes for each mouse (**Additional Fig. 6** and **Additional Fig. 7**, below). This data also shows clearly that > 90% of Confetti⁺ microglia in the contralateral hemisphere are singlets (not part of a clone). The percentage of singlets decreases significantly in the stroke region and clones emerge as early as two days after stroke. In the revised version of the manuscript, an additional analysis was incorporated that utilizes the Ki-67 proliferation marker to demonstrate that microglia proliferate and expand clonally two days after MCAo (**Supplementary Fig. 2**). Overall, we therefore believe that the current study clearly demonstrates that microglia undergo clonal expansion after MCAo.

It is a valid question as to whether clonal expansion of microglia after stroke is detrimental or beneficial. A number of prior studies have sought to examine the impact of microglial responses in experimental models of stroke. The classical approach is to deplete microglia prior to stroke induction. In general, such approaches worsen the effects of stroke, leading to larger infarct volumes and poorer functional outcomes²⁴⁻²⁷. However, it has also been observed that microglia depletion may improve stroke outcome²⁸. The reason behind such discrepancy could be the differences in the applied stroke models, the time point of outcome evaluation, or the microglia depletion paradigm. Another explanation could be that microglia play a complex role after brain ischemia with detrimental and protective mechanisms. Some studies aimed to intervene with microglial proliferation without depleting microglia. Similar to depletion, a general inhibition of microglia proliferation also results in a negative outcome^{29,30}. However, a very recent study has shown that a microglia-specific knockout of HDAC3 specifically inhibits the proliferation of proinflammatory microglia, here defined as CD16⁺ microglia, without affecting anti-inflammatory Arg1⁺ microglia, resulting in improved stroke outcome at 3 and 35 days after stroke³¹. Microglia-specific knockout of HDAC3 closes accessible regions enriched with PU.1 motifs. Induced overexpression of PU.1 partially reversed the protective effect of the knockout. The motivation behind our study goes in a similar direction. We do not think that clonal expansion after stroke should be considered protective or detrimental in general. Rather, our objective

was to investigate the contribution of clonality to the functional heterogeneity of microglia. We have shown this in **Fig. 4** of the revised manuscript. In addition, our study introduces microglial clonal expansion to the field of stroke research and establishes its dynamics following 30-minute MCAo. This provides future studies with valuable information for designing interventional paradigms. Furthermore, the description of microglial cell-cell interactions after stroke illustrates a previously unexplored behavior that merits further investigation. In the final stages of this project, we evaluated potential interventions targeting microglial clonal expansion. However, fitting these experiments into the time frame of the project seemed to lead to premature results. Therefore, we have decided to share the project in its current state with the community and to conduct interventional studies in the future. We hope that the reviewers can still appreciate the value of this work to the stroke and microglia community. We have revised the introduction and discussion to improve the placement of this study in the stroke literature.

1a) Clonal expansion implies that daughter cells from recent cell divisions are more likely to proliferate than cells that have not recently undergone cell division. Given that the authors include computational modeling approaches in their study, it feels like they could have leveraged additional modeling approaches to investigate this. For example – if you know how much the overall microglial density increased by week 2 and you can estimate the number of microglial cell division events that would be needed to achieve that density increase – it seems that you should be able to model expected clone sizes if the probability of undergoing proliferation at any given time is equal for all microglia in the tissue (purely stochastic proliferation) or if microglia that recently experienced cell division are more likely to experience additional cell divisions (daughter cells have higher probability of continuing to proliferate – i.e. bias towards clonal expansion). This type of modeling can be used to predict whether the observed clone size of 6-8 cells during weeks 2 and 4 could only be achieved with a higher proliferative capacity of daughter cells or whether it is simply what would occur as a result of stochastic microglial proliferation throughout the tissue.

Response:

The reviewer raises the interesting question of how the proliferation history of a particular cell affects its future proliferation probability. It is evident that microglial cells in the stroke region exhibit a higher proliferation probability compared to their contralateral counterparts. With the help of the new Ki-67 analysis, we can now also show that the probability of microglial proliferation in the stroke region is highest two days after MCAo (**Supplementary Fig. 2b**). The data in **Fig. 1** and **Fig. 2** show that the number of Confetti⁺ cells and the number of clones peak at two weeks after stroke and gradually go back to baseline afterward. This suggests that the probability of proliferation is adjusted for each cell over time. Two days after MCAo represents the period of acute to subacute damage, i.e., the phase when acute cell death, alterations in gene expression, and modifications in the tissue environment, accompanied by a cytokine storm, reach their zenith. During this phase, acute damage processes, quasi “environmental cues”, exert a predominant influence. Later on, the clonal expansion of microglia decreases in the absence of such environmental cues, regardless of the proliferation history of the cells. However, this does not rule out the possibility that proliferation history has a potential influence on the probability of proliferation as long as environmental cues are present. In an attempt to answer this question, we calculated the conditional probability of proliferation (being Ki-67⁺) given that the cell is a singlet, belongs to a small clone (2-5 cells), or belongs to a large clone (> 5 cells) (see figure below). The conditional probability of proliferation was found to be greatest for cells belonging to small clones, with singlets exhibiting the second-highest probability at 2 days following MCAo ($p = 0.494$ or $p = 0.346$, respectively). This probability decreases one week after MCAo and shows slight differences between singlets, cells in small clones, and cells in large clones, $p = 0.166, 0.206, 0.223$, respectively. After two weeks, the conditional probabilities were $p = 0.047, 0.07, 0.05$, in the same order. These data

suggest that cells in clones have a consistently higher probability to proliferate than singlets at all time points. However, this effect is most pronounced two days after MCAo, suggesting that the time effect, i.e., environmental cues, is more prominent than the proliferation history. This all supports that

microglial clonal expansion is a controlled physiological response to ischemic injury and that a fixed probability of proliferation based on proliferation history is not to be expected.

1b) In the discussion, they describe how clonal expansion in the context of cancer involves a subset of cells gaining a competitive advantage over other cells and, hence, they expand to a greater degree. Throughout the manuscript, they seem to imply that this is what is occurring with microglial clones in the context of stroke, but additional experiments would be needed to show that expansion with a competitive advantage is occurring in microglial clones. For example, if we assume there is no bias in the original tamoxifen induced labeling of the microglia, then labeled cells with no competitive advantage for expansion should remain as individually labeled cells or clones of 2. Cells that possess some sort of competitive advantage should proliferate more dramatically, yielding larger clones. This would result in a bimodal distribution of clone sizes throughout the stroke tissue – tiny clones (1-2 cells = no competitive advantage) and large clones (8+ cells = some sort of competitive advantage). Can the authors show histograms of clone size from individual mice and reveal whether the distribution of clone size is normal or bimodal?

Response:

We thank the reviewer for raising this point. We assume that the clonal expansion of microglia is selectively restricted to the ischemic region. In this sense, microglial cells within the ischemic region have a proliferation advantage over cells that are not exposed to ischemic conditions. However, this does not imply that specific clones acquire a competitive advantage over others and dominate the ischemic region. As requested, we show histograms of clone size for all mice in the ischemic region and the contralateral counterpart (see figures below). It is clear that the histograms do not show bimodal or normal distributions, but rather that larger clones are much less frequent than smaller clones. Previous studies have shown a similar distribution of clone size frequencies in other contexts of clonal expansion^{15,32,33}. On the contralateral side, 90-100% of the cells are singlets. The remaining percentage are mostly small clones of two cells, while larger clones are rare. In the stroke hemisphere, we observe a decrease in the percentage of singlets and a thicker and longer tail of the distribution starting two days after MCAo. These changes increase over time and are most pronounced between two to four weeks after MCAo, and then gradually decrease.

Additional Fig. 6 Histograms of clone sizes in the ischemic dorsolateral striatum from individual mice from all time points. Singlets are represented in light red color. Here it becomes easily apparent that the data do not follow a bimodal or normal distribution. Please note that the tails of some of these histograms extend beyond 15 cells. However, such events have a very low frequency and are difficult to visualize when plotting histograms from all mice.

Additional Fig. 7 Histograms of clone sizes in the contralateral striatum of individual mice from all time points. Singlets are represented in light blue color. The data show that in almost all mice Confetti⁺ cells are present as singlets (90-100%) with very few occasional small clones of two to three cells.

1c) In the discussion, the authors also describe clonal expansion of lymphocytes that have recognized their specific antigen, and the functional significance of this cellular behavior is clear. In contrast, the potential functional significance of any microglial clonal expansion in the context of stroke remains unclear. Is there any evidence that daughter cells that have undergone multiple, repetitive proliferative events are functionally different than the “older” cells in the tissue that have not experienced proliferation? This would be a super important finding for the field. Is the gene expression of cells that have experienced multiple proliferative events (members of clones containing 6-8+ cells) different than gene expression of individually labeled microglia (no proliferation) or 2 cells clones (minimal proliferation)? This could be investigated with in situ hybridization and a focus on key microglial inflammatory factors, trophic factors, phagocytosis-relevant genes or genes known to be involved in microglial-neuron or microglial-vascular interactions. One could also stain for CD68 or Clec7a or P2RY12 or similar and show whether all daughter cells in a large clone show signs of being functionally similar to one another and functionally distinct from surrounding microglia. With the current data – we have no insight into whether clonal expansion of microglia in the context of stroke is beneficial or detrimental.

Response:

We thank the reviewer for the comment and the suggestion. We tried to follow the recommendation and used immunohistochemical labeling of Clec7a, P2RY12, and CD68. Since we have observed the highest heterogeneity in the electrophysiological profiles at eight weeks after MCAo, we also attempted corresponding staining at this time point. Unfortunately, such analysis turned out to be extremely challenging. We would like to clarify that with the four Confetti markers, plus Hoechst as a nuclear marker, only one additional imaging channel remains for one additional marker in the far-red spectrum. Assigning microglial phenotypes based on only one of these additional markers is not likely to yield comprehensive results. Furthermore, we realized that none of the three markers resulted in a simple binary labeling pattern of positive and negative cells, but rather that cells exhibited a range of fluorescence intensities (**Additional Fig. 8**). We made several attempts to accurately measure the mean fluorescence intensity of individual microglial cells for each of the three markers. However, scar formation in stroke tissue complicates the process of obtaining homogeneous staining. Furthermore, the high autofluorescence of the tissue at that time point leads to additional noise in the measurement (see figure below).

Additional Fig. 8 Immunohistochemical labeling of Confetti⁺ microglia eight weeks after MCAo with microglial markers: CLEC7A, P2RY12, and CD68. Scale bars: 30 μ m.

2) *Important methodological information is missing.*

2a) *The text of the results should describe the sex and age of the mice used for the study. Is it the case that only female mice were used in this study as the methods seem to indicate? What is the justification for that choice? What is the age of the mice when they are given tamoxifen? Stroke is predominantly a brain injury that impacts aged / aging individuals. While this reviewer appreciates that it would be impractical to carry out this study in aged microfetti mice, it is nonetheless critical to be transparent about whether these studies are being carried out in young adult vs. adult vs. middle aged mice. Older is clearly better for modeling the human stroke condition – so the authors should also include in the discussion a consideration of how the aging process may impact microglial responses to stroke.*

Response:

We thank the reviewer for raising this point. As indicated in the method section, we used female mice for the experiments. The reason for this was that, when establishing this technically very demanding project for us with a wide range of sophisticated methods, we did not want to risk any methodological deviations from earlier work in which the Microfetti mouse was established. Therefore, we based our work on the study that used only female mice to investigate the clonal expansion of microglia in the facial nerve axotomy model⁴. For this revision and as a proof of principle, we conducted the same experiment on male Microfetti mice and sacrificed them one week after MCAo, obtaining similar

results. The additional figure below shows how microglia in the stroke region in male mice expanded clonally, which was confirmed by Monte Carlo simulation and DBSCAN cluster analysis.

All experiments were conducted in young adult mice, 7-13 weeks old at the time of tamoxifen injection. We have specified the sex and the age of the mice in the methods section (lines 424-426). We absolutely agree with the reviewer that the influence of age on microglial properties such as proliferation and electrophysiological profiles is very interesting and important, especially in a disease that tends to affect older age groups, such as ischemic stroke. However, since our study did not include older mice, we cannot provide a detailed answer ourselves. This will be part of future studies, especially since the literature seems to be very heterogeneous³⁴, e.g. regarding the influence of aging processes on microglial proliferation. We have added the discussion in line 409 accordingly.

Additional Fig. 9 Clonal expansion of microglia in male Microfetti mice one week after MCAo. **a** Representative microscopy images of the ischemic dorsolateral striatum with the four fluorescent Confetti proteins and an overlay with Iba-1 immunohistochemical labeling. **b** Monte Carlo simulations confirm clonal expansion in the ischemic dorsolateral striatal region, but not in the contralateral counterpart. **c** Quantification of the number of Confetti⁺ microglia shows a statistically significant increase compared to the contralateral side. The results of the DBSCAN analysis to quantify the number and size of clones after MCAo show a statistically significant increase. Statistical analysis with paired t-test, **P < 0.01. Number of animals N = 6.

2b) How are the authors defining regions of the dorsolateral striatum that are impacted by the stroke for acquiring images? The methods state that MRI analysis was used to confirm induction of stroke (this should also be described in the results section and ideally they should also include some sort of quantification of stroke severity to show how much animal-to-animal variation is present). However, once brain sections are prepared for confocal imaging and analysis, is a specific region of the dorsolateral striatum simply assumed to be impacted by stroke or do they use some type of staining in alternating brain sections to define the extent of stroke? It would be critical to know whether any cell-to-cell or clone-to-clone variation they are seeing is related to being fully within brain tissue impacted by stroke as opposed to being located at the boundaries of brain tissue impacted by stroke. For example, are the largest clones typically found within the center of tissue impacted by stroke while smaller clones are at the edges of tissue impacted by stroke?

Response:

As described in the methods section, we performed an MRI scan 72h after MCAo to confirm the induction of stroke. We are well familiar with the predicted infarct volume and lesion topology and their variations in the MCAo model that we have used for more than 20 years. The 30-minute MCAo model produces a small stroke lesion with a mean volume of 14.6 mm³ (95% CI, 11.6–17.6) as assessed by T2-weighted MRI measurement at 24h after 30min MCAo³⁵. By registering MRI images of 114 animals, the same study from our department showed that the infarct was confined to the caudoputamen in the majority of animals. However, some mice acquire a larger stroke involving nearby cortical or hippocampal regions. In all our studies, including the current one, we perform our analyses on the dorsolateral caudoputamen between Bregma levels 0.145-0.545 (Allen Brain Atlas), which is certainly involved after stroke induction. In the current study, the sections for histological preparations were also labeled for Iba-1 to visualize microglial activation and to confirm that we were imaging within the core of the stroke independent of the Confetti signal.

The figure below shows a representative T2-weighted MRI at Bregma 0.145 from one of the mice included in this study showing a typical 30-minute MCAo lesion. Also shown are the corresponding microscopic images of Iba-1 immunohistochemistry from the same animal at the same level in the ipsilateral and contralateral caudoputamen. The evaluation of the infarct volumes based on MRI images confirms that the cohort of this study is largely homogeneous. This analysis shows that the cohort is very similar in terms of lesion volume (mean = 15.02 ± 2.52 mm³) (median = 10.26 mm³[IQR: 7.50-13.92]). The mice outside this range show variable cortical or hippocampal involvement. In this study, we chose not to exclude these mice because there was no statistically significant correlation between the lesion volume at 72h and the density of the Confetti⁺ microglia at the imaged ROI in the caudoputamen. This supports that the imaged region was at the core of the infarct with no or little impact on the absolute volume of the stroke. We have added the MRI image and lesion volume analysis as **Supplementary Fig. 1c**. We have also reported this in the results section (lines 109-111) and specified our region of interest and how slices were selected for analysis in more detail in the methods section (lines 453-455 and 536-540).

Additional Fig. 10 Analysis of lesion volumes based on MRI measurement

a Representative T2-weighted MRI image at 72h after MCAo show a typical stroke lesion in the dorsolateral striatum (caudoputamen). Scale bar: 1 mm. **b** Confocal microscopic images of Iba-1 immunohistochemistry from the same mouse at the same level in the ischemic dorsolateral caudoputamen and the corresponding contralateral region. Scale bar: 100 μm . **c** Infarct volumes based on MR-imaging. Data are presented as median \pm IQR. $N = 33$ (Out of 43 mice that were included for the analyses in Fig. 1 and Fig. 2. Eight mice sacrificed at 2 days after MCAo did not receive an MRI measurement. Two mice had to be excluded because of insufficient MR-image quality for accurate measurement of lesion volume.) **d** Pearson correlation of lesion volume measured by MRI and the density of Confetti⁺ microglia in the dorsolateral caudoputamen showed no statistically significant correlation.

3) Quantification needs to be included for the analysis of “interactions” between microglia in the same or different clones. What is the frequency of process to soma, process to process, soma to soma, and entangled interactions? Is there a relationship between these interactions and morphology of the cells involved? Is there a relationship between these interactions and proximity to stroke core versus edges of tissue impacted by stroke? Moreover, the data don’t actually show that microglia from the same or different clones establish “connections” with one another (lines 375-381). Showing clearly that cell-cell “contact” occurred requires super resolution imaging approaches (for example, STED or expansion microscopy) and ideally electron microscopy. Without these sorts of additional experiments, the language of manuscript should be modified to tone down this claim and just say that there are potential or putative contacts between microglia.

Response:

We thank the reviewer for raising these important questions. During the revision process, we tried to address as many of them as possible within the revision time frame. First, our initial goal was to confirm the existence of microglia-microglia contacts using a different technique. Therefore, we used another super-resolution microscopy technique, stimulated emission depletion (STED) microscopy. We added this data as **Fig. 6b**. In fact, we observed the same types of interactions that we reported in

the original manuscript. In all these cases, microglial cells show clear interaction points. In many cases, the interaction site on one cell takes a shape complementary to its counterpart. In our opinion, these additional and new investigations prove that microglia-microglia contacts do occur. However, we also pursued the possibility of performing electron microscopy measurements on the brains of Microfetti mice after stroke. We consulted with two core electron microscopy facilities in Berlin and both confirmed that a measurement combining electron microscopy and confocal laser microscopy specifically in brain slices from *in vivo* material is technically not feasible without compromising resolution or fluorescence signal. One possible approach was to use cryoCLEM microscopy, which allows the combination of light and electron microscopy measurements on a single sample at high resolution, even allowing the collection of single-particle data. However, such a system is optimized to conduct measurements on a single-layer cell culture, and we were not advised to attempt such measurements on brain slices. The alternative option was to perform serial block-face scanning electron microscopy. This technique allows three-dimensional scanning of a tissue block at high resolution. However, correlative light microscopy measurement is not possible because the tissue block is destroyed during the imaging process and the fixation protocol eliminates any endogenous fluorescent signal. Furthermore, the data acquisition and the 3D reconstruction of such an image will require about six months of work. Therefore, the experts at the two core facilities recommended that we focus on light microscopy solutions because they are more appropriate for our sample and scientific questions.

Interestingly, a thorough literature search revealed that microglial cell-cell interactions in an experimental stroke model have been previously reported using electron microscopy³⁶. In this study, the interactions were attributed to phenomena such as cell fusion. However, our data cannot account for cell fusion. In cell fusion, two cells merge into a single object or entity with cellular membrane openings. Such a process would result in mixing of the cytoplasmic fluorescent proteins (YFP and RFP), making the fused entity positive for both markers. We did not observe such fusion behavior. On the contrary, the cell membranes of the interacting cells remain intact. With this in mind, we would like to reiterate the novelty of our results, especially after revision, and their importance for the microglial community. We have expanded the discussion of this point in lines 369-395.

In addition, we used another microscopic technique that confirms these interactions and also provides insight into the temporal dynamics of these microglia-microglia interactions. To this end, we performed live cell imaging on acute brain slices from Microfetti mice eight weeks after MCAo using an advanced fast-spinning confocal microscope equipped with high-quality silicon objectives. We present these new results in **Fig. 6c, Supplementary Fig. 7, and Supplementary Videos 2, 3, 4, and 5**). In these experiments, we performed 3D multichannel recordings of interacting cells for 10-15 minutes. These data show, in addition to the same interaction styles described above, how dynamic these interactions are. It also shows that two microglial cells can interact simultaneously with each other at different interaction points. It was also evident that one microglial cell can interact with several other cells displaying different Confetti labels (YFP or RFP). Similar to imaging on fixed slices, we did not observe any flow of cytoplasm between the interacting cells, suggesting the absence of membrane openings or fusion between interacting cells. However, we also observed that interacting cells, as shown in **Supplementary Video 4**, move together in a way that suggests membrane-membrane connections. Confirming of such connections would indeed require high-resolution electron microscopy measurements. We hope that these videos can give the reviewer an idea of how heterogeneous these interactions are and how challenging it would be to comprehensively quantify such interactions. As mentioned by the reviewer, visualization of these interactions requires high-resolution microscopy, and it would be extremely demanding to perform such measurements over the entire stroke region. Here, expansion microscopy especially in combination with light sheet

microscopy, may be an optimal approach³⁷. The establishment and comprehensive evaluation of such a method is therefore the subject of a follow-up project. We would like to point out, that we were able to identify an additional property of microglia that is characterized by significant heterogeneity, manifesting as microglia-microglia interactions. This finding complements and reinforces the other substantive statements of our project. We believe that this information should be shared with the microglia community.

4) *Few labs are equipped to carry out electrophysiological analyses of microglia, making these experiments a valuable addition to the literature. However, the field still doesn't really know what the functional significance of microglial membrane currents is and the present study doesn't really provide additional insight beyond describing observed membrane currents out to 8 weeks.*

Response:

We thank the reviewer for pointing out the value of our work, and we strongly agree that further studies, especially *in vivo* studies, are required to understand the functional significance of the microglial membrane currents. We would like to emphasize that the aim of the current study was not to provide an in-depth analysis of the microglial membrane currents and their functional significance, but to use patch-clamping as a functional readout at the single cell level in brain slices. As the reviewer points out, previous electrophysiological studies have focused on acute and subacute phases after stroke. Our data show several interesting findings from the resolution phase (8 weeks), including the higher heterogeneity of electrophysiological profiles in contrast to the acute phase, the recovery of homeostatic membrane properties in some microglial cells, and the difference in the outward current dynamics between the acute and the resolution phases. Most importantly, our patch-clamp experiments allowed us to investigate the effects of clonality and microenvironment on microglial function, as shown in the new **Fig. 4** of the revised manuscript. We believe that these experiments provide novel insights that are lacking in the literature and merit reporting.

Is there any relationship between size of the clone and the current profiles that are detected? This could suggest that there is a relationship between degree of recent proliferative events and membrane properties.

Response:

During patch clamp experiments, a rough subjective estimate of clone size can be made. Unfortunately, it is not possible to obtain an accurate measurement of clone size for the following technical reasons. First, we patch cells using a 63x microscope objective, which provides a narrow field of view in the x and y dimensions and a narrow depth of focus in the z dimension. This makes it difficult to visualize all cells belonging to the same clone in one image. Second, it was not possible to apply computational spatial clustering algorithms during the patching process. Such analyses were performed on fixed slices and required sophisticated imaging, segmentation and computational analyses. Third, the patch-clamp experiments are time-critical and all recordings should be acquired within three hours after acute slice preparation.

A number of older reports have demonstrated that K^+ channel activity regulates various microglial functions, including proliferation, migration, and chemotaxis^{38,39}. Indeed, the increase in K^+ channel activity results in a hyperpolarization of microglial cells, thereby stabilizing the membrane potential at a negative value. However, it is important to note that none of the K^+ channels is exclusively activated during proliferation. Furthermore, these studies were conducted in cell cultures. Future *in vivo* studies with specific targeting of these channels are needed to confirm their specific role in microglial proliferation.

Is membrane capacitance correlated with inward or outward currents, which might suggest relationships between membrane properties and morphological properties?

Response:

A similar question was also asked by reviewer 2. Please kindly refer to our comments on reviewer 2 (question 3) above.

For the microglial cells recorded in the contralateral hemisphere – were they recorded from all the different time points and then just pooled? Were there any subtle changes in number of cells that have the moderate inward conductance in the contralateral hemisphere across time points? There are almost certainly changes in the CSF milieu after injury and patterns of circuit activity beyond the stroke lesion site will be altered. Hence, it would almost seem a bit surprising if there was NO change in the contralateral microglia after stroke. These types of data could provide insight into how much microglial electrophysiological properties are driven by immediate micro-environment versus more systemic status of the CNS.

Response:

We would like to thank the reviewer for raising this important point. Global effects after focal ischemic stroke have indeed been described⁴⁰. In this context, the unchanged or homeostatic morphology of microglia in the contralateral hemisphere seems surprising, but it is a fact. Consistent with this, in our previous and current work, we never observed an effect of stroke on the electrophysiological profile of microglia in the contralateral hemisphere after 30-minutes MCAo. The figure below shows the IV-curves from all contralateral cells included in this study, in addition to recordings from the microglia of naïve MacGreen (Csf1r-EGFP) or Cx3cr1-GFP mice. We did not observe significant differences in the presence of moderate inward currents across time points or when compared to naïve mice. Furthermore, a comparison of the basic electrophysiological properties of the four groups showed no statistically significant difference. Therefore, cells from the contralateral side from all time points were all pooled into one control group. We clarified this in the revised manuscript in lines 187-188.

Additional Fig. 11 Electrophysiological properties and types of currents in control microglia

a IV-curves of voltage clamp recordings from microglia from naïve MacGreen/Cx3Cr1 mice and from the contralateral hemispheres at 2 days, 7 days, and 8 weeks after 30-minutes MCAO. Recordings were performed at a holding potential of -70 mV. Each curve represents one cell. **b** The above-shown cells were compared regarding their basic electrophysiological properties similar to the analysis in Fig. 3c (membrane resistance R_m , capacity C_m , reversal potential V_{rev} , specific outward conductance G_{out} [from 0 to -20 mV] and specific inward conductance G_{in} [from -120 to -100 mV]). Data are presented as median and interquartile range. The Kruskal-Wallis test shows no statistically significant difference between the groups. Therefore, the cells from the contralateral hemisphere were pooled and used as a control group.

Minor concerns:

1) For the t-SNE based analysis of morphology, did the authors examine which of the 28 extracted morphological feature are highly correlated? Ideally they should show a correlation heatmap. Did they consider eliminating some highly correlated features before performing dimensionality reduction clustering? Otherwise, if they are including numerous morphological features that essentially capture the same thing about cell morphology, that morphological feature will be accorded undue weight in the subsequent clustering.

Response:

We are grateful for the insightful commentary provided by the reviewer, which has led to the enhancement of our study. In response to the request, we present the requested correlation heatmap (see figure below). The analysis revealed a high degree of collinearity among numerous variables. Consequently, we opted for an alternative analytical approach, wherein we selected the minimum number of the most salient variables. These variables exhibited the most significant differences between microglia from the contralateral side and microglia in the ischemic tissue, as well as between microglia in the ischemic tissue at the different time points (shown in Fig. 5b). We followed this with proper scaling of the data and conducted a UMAP analysis for dimensional reductionism, a method which is preferable to the tSNE analysis. UMAP is a nonlinear analysis method that is less sensitive to collinearity than other methods, such as PCA, and more reproducible than tSNE plots. This resulted in a plot that resembles but is superior to the original tSNE plot (Fig. 5c and Supplementary Fig. 5c). Similar to the original plot, a clear distinction is observed between highly ramified cells and other cells. In addition, the new plot demonstrates an improved resolution in differentiating cells from ischemic lesions at different time points. Here, cells from two days after stroke occupy the upper right pole of the plot, while cells from eight weeks occupy the lower right pole, and cells from one week span over

that part of the plot. This gradient was already present in the original tSNE plot, however, the current UMAP provides even better visualization.

Additional Fig. 12 Spearman correlation heatmap for all morphological parameters.

Additional Fig. 13 Spearman correlation heatmap for the finally selected morphological parameters.

2) The idea that microglia within one clone are “functionally homogeneous” (line 302 and lines 359, 360) seems like a stretch. The authors have shown that two recorded cells within one clone show

similar membrane currents. However more information would be needed to say that they are functionally homogeneous. For example, are the cells that are within one clone also similar in terms of gene expression, cell process motility, phagocytic capacity? Without these sorts of additional experiments, the language of manuscript should be modified to tone down this claim.

Response:

We apologize for this overestimation. We agree that a claim of functional homogeneity would require additional features that we have not evaluated. We revised the manuscript and used the term “functional correlation” instead in line 313.

3) For readers who are not in the stroke field, it would be helpful to provide a few sentences in the text of the results describing the 30min MCAO model and comparing it to other stroke models (relative severity at the tissue level – time course of any necrosis, blood brain barrier compromise and infiltration of peripheral cells; also relative severity at the behavioral level – does this model result in behavioral deficits comparable to those experienced by human stroke patients). This information will also hopefully allow the authors to justify why they chose this particular stroke model for their study. I may have missed it, but I don't believe they gave any rationale for using this model.

Response:

We thank the reviewer for this valuable advice. There are several preclinical (experimental) models of stroke that reflect different clinical scenarios⁴¹⁻⁴³. Clinically, ischemic strokes affecting the territory of the middle cerebral artery (MCA) are most common⁴⁴. Consequently, MCA occlusion (MCAo) is the most commonly used technique for inducing a focal ischemic brain lesion in preclinical studies. Over time, various MCAo techniques have been developed⁴⁵⁻⁵¹. Each of these models has specific advantages and disadvantages and should be used in accordance with the scientific question being addressed. In our laboratory, we have more than 25 years of experience with a stroke model in which the proximal origin of the middle cerebral artery can be (transiently) occluded with an intraluminally applied silicone-coated monofilament⁵². This intraluminal, so-called filament model has the advantage of allowing the surgeon to occlude the MCA via an endovascular approach, making it relatively non-invasive compared to models that require craniectomy (such as the distal MCAo model). The duration of focal brain ischemia can be easily controlled by removing the filament, which allows reperfusion of the MCA territory (transient MCAo). The infarct size in the filament MCAo model, and therefore stroke severity, is directly related to the duration of the occlusion. Permanent occlusion of the MCA, where the filament is not removed, resembles malignant stroke in patients without recanalization or thrombolytic therapy. Permanent occlusion of the MCA results in large hemispheric stroke, marked brain edema, and a significant lesion expansion due to delayed ischemic cell death of all brain cell types. The permanent proximal MCAo model does not allow the study of chronic stroke outcomes because the animals typically do not recover. Reperfusion occurs in filament transient MCAo models and resembles the human situation in the case of spontaneous recanalization or in the case of successful therapy, e.g. endovascular therapy/thrombectomy or systemic thrombolysis. 60-90 minutes of MCAo produces larger infarcts involving the lateral striatum and the frontoparietal cortex and causes pannecrosis. In contrast, 30-minute MCAo leads to selective neuronal cell death that is restricted to the striatum, with glial cell types not dying. This is an essential prerequisite for the project's objectives. Furthermore, after 30-minute MCAo, the animals initially exhibit typical stroke symptoms, which can be quantified using behavioral tests, but recover functionally, thus enabling long-term survival of the mice^{41,53,54}. In our opinion, this is also an advantage over distal MCAo, which does not really lead to quantifiable neurological deficits that can be measured with standard behavioral tests, although cortical structures are affected and this model usually does not allow

reperfusion, i.e. it corresponds to permanent occlusion. Pharmacological stroke models (photothrombosis or vasoconstriction by endothelin) are not included in our projects.

The extent of blood-brain barrier dysfunction, non-neuronal brain cell responses, leucocyte infiltration, and behavioral outcomes for the 30-minutes MCAo/reperfusion model have been previously reported by us^{22,41,54-71}. As the detailed explanation above is beyond the scope of this publication, we have included the following sentences in the revised manuscript (line 102-107):

“We have chosen the proximal intraluminal 30 min MCAo/reperfusion model because of high reproducibility, robust induction of microglia responses, and functional recovery - consequently, the possibility to study long-term sequelae of cerebral ischemia. This MCAo model mimics human large vessel occlusion with thrombectomy. The lesions are characterized by selective neuronal cell death limited to the ipsilateral striatum, resulting in a defined impairment of sensorimotor function¹⁸.”

4) Can the authors use the discussion to relate the findings from their various types of analysis? For example, how does the extent and time course of recovery of microglial morphological complexity after stroke compare to extent and time course of recovery of microglial electrophysiological properties after stroke? Are they similar? Is one faster? Are there relationships between typical morphological characteristics of cells and the size of the clone they are from?

Response:

We thank the reviewer for this suggestion. We tried to improve the discussion and create more coherence between the different analyses, as suggested by the reviewers. (See changes in lines 350-368)

5) Fig 1 – something looks very different about the 8W and 12W images – different scale? Is there more background fluorescence? Is there cellular debris?

Response:

At the chronic time points (8 weeks and 12 weeks), we do indeed see increased background signal, likely due to lipofuscin-autofluorescence. We have pointed this out in the revised legend of **Fig. 1**. All images within an analysis were taken with the same imaging setup and scale. The matching scale can be confirmed from the general anatomy of the region, where the corpus callosum occupies a matching area in the upper left corner of all images in **Fig. 1d**.

6) Line 282 is likely a typo – probably should be “wrap” rather than “warp” around each other

Response:

We thank the reviewer for pointing out this typo. We have corrected it accordingly.

7) Figure 4c – need to put “cell 1” and “cell 2” above the two examples of responses to hyperpolarization / depolarization to aid in reader understanding of the figure

Response:

We thank the reviewer for this suggestion. We incorporated this into the figure, resulting in further improvement.

I do not have expertise in coding and hope that other reviewers can comment on this aspect of the study.

Response:

We thank the reviewer for providing suggestions that improved our analyses, and we encourage reviewers to check our code, which we have provided along with demo data for testing. The code will also be made publicly available upon acceptance of the manuscript.

References

1. Yenari, M. A., Kauppinen, T. M. & Swanson, R. A. Microglial Activation in Stroke: Therapeutic Targets. *Neurotherapeutics* **7**, 378–391 (2010).
2. Qin, C. *et al.* Dual Functions of Microglia in Ischemic Stroke. *Neurosci. Bull.* **35**, 921–933 (2019).
3. Rawlinson, C., Jenkins, S., Thei, L., Dallas, M. L. & Chen, R. Post-Ischaemic Immunological Response in the Brain: Targeting Microglia in Ischaemic Stroke Therapy. *Brain Sciences* **10**, 159 (2020).
4. Tay, T. L. *et al.* A new fate mapping system reveals context-dependent random or clonal expansion of microglia. *Nat Neurosci* **20**, 793–803 (2017).
5. Zhan, L. *et al.* Proximal recolonization by self-renewing microglia re-establishes microglial homeostasis in the adult mouse brain. *PLOS Biology* **17**, e3000134 (2019).
6. Jordão, M. J. C. *et al.* Single-cell profiling identifies myeloid cell subsets with distinct fates during neuroinflammation. *Science* **363**, eaat7554 (2019).
7. Masuda, T. *et al.* Specification of CNS macrophage subsets occurs postnatally in defined niches. *Nature* **604**, 740–748 (2022).
8. Cabeza-Cabrerizo, M. *et al.* Tissue clonality of dendritic cell subsets and emergency DCpoiesis revealed by multicolor fate mapping of DC progenitors. *Science Immunology* **4**, eaaw1941 (2019).
9. Tas, J. M. J. *et al.* Visualizing antibody affinity maturation in germinal centers. *Science* **351**, 1048–1054 (2016).
10. Snippert, H. J. *et al.* Intestinal Crypt Homeostasis Results from Neutral Competition between Symmetrically Dividing Lgr5 Stem Cells. *Cell* **143**, 134–144 (2010).
11. Schepers, A. G. *et al.* Lineage Tracing Reveals Lgr5+ Stem Cell Activity in Mouse Intestinal Adenomas. *Science* **337**, 730–735 (2012).

12. Lasrado, R. *et al.* Lineage-dependent spatial and functional organization of the mammalian enteric nervous system. *Science* **356**, 722–726 (2017).
13. Chappell, J. *et al.* Extensive Proliferation of a Subset of Differentiated, yet Plastic, Medial Vascular Smooth Muscle Cells Contributes to Neointimal Formation in Mouse Injury and Atherosclerosis Models. *Circulation Research* **119**, 1313–1323 (2016).
14. Weber, K. *et al.* RGB marking facilitates multicolor clonal cell tracking. *Nat Med* **17**, 504–509 (2011).
15. Barry-Carroll, L. *et al.* Microglia colonize the developing brain by clonal expansion of highly proliferative progenitors, following allometric scaling. *Cell Reports* **42**, 112425 (2023).
16. Young, A. *et al.* Targeted evolution of adeno-associated virus capsids for systemic transgene delivery to microglia and tissue-resident macrophages. *Proceedings of the National Academy of Sciences* **120**, e2302997120 (2023).
17. Stamataki, M., Rissiek, B., Magnus, T. & Körbelin, J. Microglia targeting by adeno-associated viral vectors. *Front. Immunol.* **15**, (2024).
18. Dumas, L., Clavreul, S., Michon, F. & Loulier, K. Multicolor strategies for investigating clonal expansion and tissue plasticity. *Cell Mol Life Sci* **79**, 141 (2022).
19. Livet, J. *et al.* Transgenic strategies for combinatorial expression of fluorescent proteins in the nervous system. *Nature* **450**, 56–62 (2007).
20. Faust, T. E. *et al.* A comparative analysis of microglial inducible Cre lines. *Cell Reports* **42**, 113031 (2023).
21. Huang, Y. *et al.* Repopulated microglia are solely derived from the proliferation of residual microglia after acute depletion. *Nat Neurosci* **21**, 530–540 (2018).
22. Kronenberg, G. *et al.* Distinguishing features of microglia- and monocyte-derived macrophages after stroke. *Acta Neuropathol* **135**, 551–568 (2018).
23. Paolicelli, R. C. *et al.* Microglia states and nomenclature: A field at its crossroads. *Neuron* **110**, 3458–3483 (2022).

24. Szalay, G. *et al.* Microglia protect against brain injury and their selective elimination dysregulates neuronal network activity after stroke. *Nat Commun* **7**, 11499 (2016).
25. Jin, W.-N. *et al.* Depletion of microglia exacerbates postischemic inflammation and brain injury. *J Cereb Blood Flow Metab* **37**, 2224–2236 (2017).
26. Marino Lee, S., Hudobenko, J., McCullough, L. D. & Chauhan, A. Microglia depletion increase brain injury after acute ischemic stroke in aged mice. *Experimental Neurology* **336**, 113530 (2021).
27. Yang, S. *et al.* TREM2-IGF1 Mediated Glucometabolic Enhancement Underlies Microglial Neuroprotective Properties During Ischemic Stroke. *Advanced Science* **11**, 2305614 (2024).
28. Li, T. *et al.* Specific depletion of resident microglia in the early stage of stroke reduces cerebral ischemic damage. *Journal of Neuroinflammation* **18**, 81 (2021).
29. Lalancette-Hébert, M., Gowing, G., Simard, A., Weng, Y. C. & Kriz, J. Selective Ablation of Proliferating Microglial Cells Exacerbates Ischemic Injury in the Brain. *J. Neurosci.* **27**, 2596–2605 (2007).
30. Hou, B. *et al.* Pharmacological Targeting of CSF1R Inhibits Microglial Proliferation and Aggravates the Progression of Cerebral Ischemic Pathology. *Front. Cell. Neurosci.* **14**, (2020).
31. Zhang, Y. *et al.* Arresting the bad seed: HDAC3 regulates proliferation of different microglia after ischemic stroke. *Science Advances* **10**, eade6900 (2024).
32. Gaimann, M. U., Nguyen, M., Desponds, J. & Mayer, A. Early life imprints the hierarchy of T cell clone sizes. *eLife* **9**, e61639 (2020).
33. Shevryev, D., Tereshchenko, V. & Kozlov, V. Immune Equilibrium Depends on the Interaction Between Recognition and Presentation Landscapes. *Front. Immunol.* **12**, (2021).
34. Antignano, I., Liu, Y., Offermann, N. & Capasso, M. Aging microglia. *Cell. Mol. Life Sci.* **80**, 126 (2023).
35. Knab, F. *et al.* Prediction of Stroke Outcome in Mice Based on Noninvasive MRI and Behavioral Testing. *Stroke* **54**, 2895–2905 (2023).

36. Ito, U., Nagasao, J., Kawakami, E. & Oyanagi, K. Fate of Disseminated Dead Neurons in the Cortical Ischemic Penumbra. *Stroke* **38**, 2577–2583 (2007).
37. Gao, R. *et al.* Cortical column and whole-brain imaging with molecular contrast and nanoscale resolution. *Science* **363**, eaau8302 (2019).
38. Stebbing, M. J., Cottee, J. M. & Rana, I. The Role of Ion Channels in Microglial Activation and Proliferation – A Complex Interplay between Ligand-Gated Ion Channels, K⁺ Channels, and Intracellular Ca²⁺. *Frontiers in Immunology* **6**, (2015).
39. Lam, D. & Schlichter, L. C. Expression and contributions of the Kir2.1 inward-rectifier K⁺ channel to proliferation, migration and chemotaxis of microglia in unstimulated and anti-inflammatory states. *Front. Cell. Neurosci.* **9**, (2015).
40. Pilipenko, V. *et al.* Focal Cerebral Ischemia Induces Global Subacute Changes in the Number of Neuroblasts and Neurons and the Angiogenic Factor Density in Mice. *Medicina (Kaunas)* **59**, 2168 (2023).
41. *Rodent Models of Stroke*. vol. 120 (Springer, New York, NY, 2016).
42. Fluri, F., Schuhmann, M. K. & Kleinschnitz, C. Animal models of ischemic stroke and their application in clinical research. *Drug Des Devel Ther* **9**, 3445–3454 (2015).
43. Liu, F. & McCullough, L. D. Middle cerebral artery occlusion model in rodents: methods and potential pitfalls. *J Biomed Biotechnol* **2011**, 464701 (2011).
44. Lyden, P. D. *et al.* The Stroke Preclinical Assessment Network: Rationale, Design, Feasibility, and Stage 1 Results. *Stroke* **53**, 1802–1812 (2022).
45. Buchan, A. M., Xue, D. & Slivka, A. A new model of temporary focal neocortical ischemia in the rat. *Stroke* **23**, 273–279 (1992).
46. Koizumi, J., Yoshida, Y., Nakazawa, T. & Ooneda, G. Experimental studies of ischemic brain edema. *Nosotchu* **8**, 1–8 (1986).
47. Longa, E. Z., Weinstein, P. R., Carlson, S. & Cummins, R. Reversible middle cerebral artery occlusion without craniectomy in rats. *Stroke* **20**, 84–91 (1989).

48. Macrae, I. M., Robinson, M. J., Graham, D. I., Reid, J. L. & McCulloch, J. Endothelin-1-induced reductions in cerebral blood flow: dose dependency, time course, and neuropathological consequences. *J Cereb Blood Flow Metab* **13**, 276–284 (1993).
49. Orset, C. *et al.* Mouse model of in situ thromboembolic stroke and reperfusion. *Stroke* **38**, 2771–2778 (2007).
50. Tamura, A., Graham, D. I., McCulloch, J. & Teasdale, G. M. Focal cerebral ischaemia in the rat: 1. Description of technique and early neuropathological consequences following middle cerebral artery occlusion. *J Cereb Blood Flow Metab* **1**, 53–60 (1981).
51. Watson, B. D., Dietrich, W. D., Busto, R., Wachtel, M. S. & Ginsberg, M. D. Induction of reproducible brain infarction by photochemically initiated thrombosis. *Ann Neurol* **17**, 497–504 (1985).
52. Endres, M. *et al.* Stroke protection by 3-hydroxy-3-methylglutaryl (HMG)-CoA reductase inhibitors mediated by endothelial nitric oxide synthase. *Proc Natl Acad Sci U S A* **95**, 8880–8885 (1998).
53. Engel, O., Kolodziej, S., Dirnagl, U. & Prinz, V. Modeling stroke in mice - middle cerebral artery occlusion with the filament model. *J Vis Exp* 2423 (2011) doi:10.3791/2423.
54. Katchanov, J. *et al.* Selective neuronal vulnerability following mild focal brain ischemia in the mouse. *Brain Pathol* **13**, 452–464 (2003).
55. Balkaya, M., Kröber, J., Gertz, K., Peruzzaro, S. & Endres, M. Characterization of long-term functional outcome in a murine model of mild brain ischemia. *J Neurosci Methods* **213**, 179–187 (2013).
56. Boujon, V. *et al.* Dual PPAR α / γ agonist aleglitazar confers stroke protection in a model of mild focal brain ischemia in mice. *J Mol Med (Berl)* **97**, 1127–1138 (2019).
57. Costa, A. *et al.* Deletion of muscarinic acetylcholine receptor 3 in microglia impacts brain ischemic injury. *Brain Behav Immun* **91**, 89–104 (2021).
58. Donath, S. *et al.* Interaction of ARC and Daxx: A Novel Endogenous Target to Preserve Motor Function and Cell Loss after Focal Brain Ischemia in Mice. *J Neurosci* **36**, 8132–8148 (2016).

59. Endres, M. *et al.* Folate deficiency increases postischemic brain injury. *Stroke* **36**, 321–325 (2005).
60. Endres, M. *et al.* Serum insulin-like growth factor I and ischemic brain injury. *Brain Res* **1185**, 328–335 (2007).
61. Gertz, K. *et al.* Essential role of interleukin-6 in post-stroke angiogenesis. *Brain* **135**, 1964–1980 (2012).
62. Gertz, K. *et al.* Partial loss of VE-cadherin improves long-term outcome and cerebral blood flow after transient brain ischemia in mice. *BMC Neurol* **16**, 144 (2016).
63. Gertz, K. *et al.* Withdrawal of statin treatment abrogates stroke protection in mice. *Stroke* **34**, 551–557 (2003).
64. Harhausen, D. *et al.* CD93/AA4.1: a novel regulator of inflammation in murine focal cerebral ischemia. *J Immunol* **184**, 6407–6417 (2010).
65. Ji, S. *et al.* Acute neuroprotection by pioglitazone after mild brain ischemia without effect on long-term outcome. *Experimental Neurology* **216**, 321–328 (2009).
66. Katchanov, J. *et al.* Mild cerebral ischemia induces loss of cyclin-dependent kinase inhibitors and activation of cell cycle machinery before delayed neuronal cell death. *J Neurosci* **21**, 5045–5053 (2001).
67. Kronenberg, G. *et al.* Exofocal dopaminergic degeneration as antidepressant target in mouse model of poststroke depression. *Biol Psychiatry* **72**, 273–281 (2012).
68. Kronenberg, G. *et al.* Folate deficiency induces neurodegeneration and brain dysfunction in mice lacking uracil DNA glycosylase. *J Neurosci* **28**, 7219–7230 (2008).
69. Kronenberg, G. *et al.* Nestin-Expressing Cells Divide and Adopt a Complex Electrophysiologic Phenotype after Transient Brain Ischemia. *J Cereb Blood Flow Metab* **25**, 1613–1624 (2005).
70. Wegner, S. *et al.* Endothelial Cell-Specific Transcriptome Reveals Signature of Chronic Stress Related to Worse Outcome After Mild Transient Brain Ischemia in Mice. *Mol Neurobiol* **57**, 1446–1458 (2020).

71. Yildirim, F. *et al.* Histone acetylation and CREB binding protein are required for neuronal resistance against ischemic injury. *PLoS One* **9**, e95465 (2014).

Response to referees:

We sincerely thank the reviewers again for their time and effort to review our current manuscript. We did our best to address the remaining issues and handle any remaining concerns. The changes in the updated manuscript are highlighted in **red**. Reviewers' comments are presented in *italic* format.

Reviewer #1

The authors have made an effort to justify their approach on the basis of the highlighted caveats on my initial review. However, the revision falls short by only providing justifications without resolving the actual issues. For example:

Points 1 and 3: issues with microfetti mice. The authors provide further justifications but this does not modify the fact that their system is not capable of distinguishing if two cells that share colour belong to the same clone or just happened to be recombined to express the same colour (1/4 chance). This is particularly poignant for RFP cells (linked to point 3), as 40% of the recombined cells express RFP hence the likelihood of two cells expressing RFP out of chance is really high.

In short, the authors cannot provide a conclusive answer to this question, based on the used method, so all the downstream analysis is heavily compounded by this issue. The fact that the system has been used in other studies does not remove this caveat. If any, it adds the same issue to those published studies.

Response:

We thank the reviewer for the feedback and appreciate the concern regarding the possibility of neighboring cells acquiring the same color by chance. However, we do not believe that the downstream analyses are as compromised by this issue as depicted by the reviewer.

As we explained in our previous response, the tamoxifen dosage used in this study results in sparse labeling of microglia, about 17% of microglial cells in the contralateral hemisphere, with 137.5 ± 7.6 μm as the average distance to the nearest neighbor of the same color. We have shown in our previous response (**Additional Fig. 7**, now **Supplementary Fig. 2**) that in most animals, > 90% of Confetti⁺ cells in the contralateral hemisphere are present as singlets, i.e., without a close neighbor sharing the same color.

Therefore, although the reviewer points out rightly to the possibility of this event, we have already shown data that this event is rare and has little impact. To clarify this point to the reader, we added to the revised manuscript (**Supplementary Fig. 2** and **Supplementary Fig. 3**), which were presented in our previous response as (**Additional Fig. 7** and **Additional Fig. 6**), respectively. Additionally, we explicitly discussed this point in the revised manuscript in lines **413-425**.

Point 5: despite the explanations, and the new data provided, the morphological, clonal, and electrophysiological data is mostly descriptive and correlational. This study lacks mechanistic evidence connecting these multiple aspects, and most importantly lacks novelty in how it's advancing our understanding of the pathophysiology of stroke.

To sum up, I'm afraid I still believe this study does not merit publication, both on the grounds of technical flaws and lack of novelty.

Response:

We thank the reviewer for the critical evaluation. Despite the limitations of our study, we believe that it provides valuable insights for the stroke and microglia research community. We have openly and lengthily discussed the strengths and weaknesses of our study and tried our best to address and clarify all issues of concern. We are very sorry that despite our extended effort, we could not convince the reviewer with our approaches. We hope that future studies will be able to build on our findings and address all remaining issues using more advanced technologies that can combine functional and anatomical readouts from the same sample on single cell level.

Reviewer #2

The authors response to the raised critique is rather extensive and covered well the concerns raised by reviewers. At this time I do not have any additional concerns and/or questions.

Response:

We sincerely thank the reviewer for the constructive feedback throughout the revision process and for contributing to improving the quality of the current manuscript.

Reviewer #3

In this revised manuscript, Kikhia and colleagues have added additional experiments and analyses that have greatly strengthened the study. In general, they have responded to most of the concerns raised by me and other reviewers. However, there is one key point that has not been fully resolved and, in my mind, is an important issue for the field. This is regarding the use of the term “clonal expansion.”

The classical, textbook definition of clonal expansion grew out of the cancer and immunology fields and involves thinking of cellular population dynamics as similar to Darwinian population dynamics of whole organisms. In the cancer field, this dates back to 1976, with Peter Nowell’s landmark perspective (PMID: 959840) arguing that individual somatic cells incur mutations that result in malignant transformation. They then proliferate and begin to outnumber healthy neighboring cells. Because all the malignant cells are daughters of an initial parent that incurred mutations, this represents a “clonal expansion” by which numerous daughter cells with some key defining attributes come to dominate the local cellular population. Publications in the cancer field continue to use the term “clonal expansion” to refer to a specific group of cells with unique, identifiable attributes that have become overrepresented in the local cell population through proliferation and a Darwinian, natural selection like process (for example PMID: 17109012 and PMID: 28912577).

The same principles are evident in Immunology, where the classic, textbook example of clonal expansion is the behavior of lymphocytes. In 1957 and 1959, David Talmage and Sir Frank Macfarlane Burnet proposed that lymphocyte function and behavior were driven by a “selection process” (PMID 13425332, and “The Clonal Selection Theory of Acquired Immunity, Vanderbilt Univ. Press, 1959). The idea was that lymphocytes stochastically express numerous diverse antigen receptors and that when a lymphocyte encounters a ligand for its particular antigen receptor, this drives a massive proliferation and differentiation of those specific cells. Again – through massive proliferation beginning with one or few parent cells, these daughter lymphocytes with a unique set of attributes come to (transiently) dominate the overall population of lymphocytes. This is the essence of clonal expansion and is the

manner in which the immunology field (at least the vast majority of the field) continues to use this term (PMID 32424244, PMID 38506411).

Other fields have used the term “clonal expansion” with the same underlying meaning that a specific subset of cells with unique attributes come to dominate the overall cell population through proliferation from very few parent cells. This includes the cardiovascular field and formation of atherosclerotic plaques (PMID: 38362345) where one of the studies the authors cite (Chappell et al 2016, PMID: 27682618) uses Microfetti to show that plaques are dominated by progeny of one or two clones. They state “In contrast to the mosaic stochastic labeling observed in the vascular wall, VSMC-derived cells within plaques were found in large monochromatic regions with little intermixing between colors.” They conclude that “extensive proliferation of a low proportion of highly plastic VSMCs results in the observed VSMC accumulation after injury and in atherosclerotic plaques.”

Hence, the essential elements of the process of clonal expansion are 1) that cells proliferate to give rise to daughter cells and 2) that a specific group of cells with some unique attribute (activated antigen receptor, malignant transformation, unusually high proliferative potential) outcompetes the other local cells and becomes a huge clone that is over-represented within the local cellular environment/cellular population. In other words, to apply the term “clonal expansion,” it’s not enough to show that individual clones increase in size/cell number. You need to show that specific clones expand RELATIVE TO OTHER CLONES and/or relative to the local cell population.

The present study and other studies that have used microfetti mice to study microglia and macrophages (Tay et. al, 2017, Zhan et. al 2019, Jordao 2019, Masuda 2022) have used this tool to show convincingly that microglia/macrophages proliferate in various contexts. This proliferation results in clones with roughly 2-25 same color daughter cells. This allows them to move beyond what is possible with BrdU labeling or single color lineage tracing to show how many cell divisions typically arise from an individual parent, to explore whether neighboring clones have distinct properties or disparate capacity to support health and function of nearby synapses/neurons etc, and to see if cells from neighboring clones intermix or interact with one another.

Microfetti can also be used to see if any clones come to be overrepresented or dominate the local environment. However, none of the studies listed above, nor the present study have not shown that any microglial clones appear to have some sort of competitive advantage over other clones or that specific types of microglial clones become over-represented and dominate the local cellular environment. Showing that clones within or near an area of pathology proliferate more than clones distant from pathology is not the same as showing that – within the pathology zone - specific clones have elevated proliferative capacity or a competitive advantage relative to other clones that are also in the pathology zone. If you haven’t shown this, then you haven’t met the criteria for claiming that clonal expansion is occurring in the context of that pathology.

In their rebuttal, the authors point to use of Monte Carlo simulation as a statistical approach to show that clonal expansion is occurring. This is also inaccurate. In the words of Tay et al, Monte Carlo simulation allowed them to “infer whether same-color microglia were related as progeny of proliferation events or were located in the same neighborhood simply due to chance.” In the words of Cabeza – Cabrerizo 2019, Monte Carlo simulation was used “to determine the probability that the observed clusters could have arisen by chance.” In other words, Monte Carlo simulation can confirm that your observed cell distribution of cells of different colors occurred as a result of proliferation of

parent cells and location of daughter cells near to the parent, rather than simply due to chance. But this does not prove that a process of “clonal expansion” – as it is classically defined – is occurring.

I strongly believe that the current manuscript makes very valuable additions to our understanding of the dynamics of microglial responses to MCAo. They show that daughter cells that are part of the same clone share key properties (e.g. potassium channel expression) that can differ from those of cells in a neighboring clone. They reveal key nuanced details about the dynamics of microglial proliferation, about some of the mechanisms that can underlie cell to cell heterogeneity during responses to MCAo, and the dynamics of returning microglia to baseline numbers during the resolution phases of responses to pathological insults. Even if none of the microglial clones have a competitive advantage and are not undergoing classical “clonal expansion,” it will be very important to understand if clones with different properties (e.g. potassium channel expression) have a greater or lesser capacity to support local synapse/neuron health and tissue repair.

But I also strongly feel that the present and recent microglial studies are using the term “clonal expansion” in a way that may confuse the reader and imply that certain cellular population dynamics similar to those observed in immunology and cancer are occurring when they are not. In my opinion, the authors should not use this term to describe their data and should instead use some sort of term like “clonal dynamics” or “clonal heterogeneity.” Of course, it will be up to the editors to make a final decision on this matter. At the very minimum, there needs to be a clear discussion of the classical definitions of clonal expansion earlier in the paper (introduction) and very clear statements that they do not have evidence that specific microglial clones exhibit any competitive advantage relative to other clones during responses to MCAo, so the clonal microglial dynamics they are observing differ from what is observed in immunology and cancer.

I apologize if this long discussion seems pedantic, as that was not my intent. This is an excellent and important study that can further do a service to the field by being very careful and judicious in use of terminology.

Response:

We sincerely thank the reviewer for the in-depth review of the literature with the aim of enhancing the accuracy of the terminology used in the current study and the field of microglia research in general. We are also grateful for the emphasis on the quality and significance of our study.

In summary, the reviewer questions the accuracy of using the term “clonal expansion” to describe the proliferation process of microglia in response to ischemic stroke. The reviewer proposes that to describe a proliferation process as “clonal expansion,” two conditions must be met:

1. A mother cell proliferates, giving rise to daughter cells with key identifiable attributes, i.e., a clone.
2. A single (or very few) clone must outcompete other clones, or cells, and dominate the local environment.

We completely agree with the reviewer that these two conditions are usually met by the classical examples of clonal expansion in immunology and cancer biology. We also agree with the reviewer that our study and recent microglia proliferation dynamics studies did not show evidence for the second criterion. Indeed, our data did not show the dominance of a single microglia clone after ischemic stroke

but rather an enlargement of numerous clones with various proliferation capacities (**Fig. 1**, **Fig.2**, **Supplementary Fig. 3**, and **Supplementary Fig. 4**). Therefore, it is reasonable to suggest the use of different terminologies to describe these dynamics. However, we would like to draw attention to the fact that the term “clonal expansion” is indeed used in other contexts that resemble the process of microglial proliferation dynamics after ischemic stroke. For example, in lineage tracing studies, similar to ours, researchers from independent groups and different fields use the term “clonal expansion” when they observe an increase in the size of a clone of cells identified genetically or with a fluorescent tag, even in the absence of single clone domination¹⁻⁴. Here, the term simply refers to the fact that there are identifiable “clones” generated from single mother cells, and these clones are increasing in size, i.e., “expanding”. Furthermore, the term “clonal expansion” is also used to describe the increase of size of multiple clones of cells that acquired driver mutations in the context of chronic inflammation and aging^{5,6} (please check ref.⁵, **Fig. 2**). In other words, there are independent studies, including ones published in *Nat. Commun.* and other *Nature* journals, that use the term clonal expansion without the second criterion being met.

That being said, there are other terminologies in the literature that can differentiate between the cases where a single (or very few) clone(s) dominates, in contrast to the case where many clones increase in size. For example, using mitochondrial lineage tracing, Wang et al.⁷ differentiate between weak clonal expansion, where many cell lineages persist, and strong clonal expansion, where a very few clones prevail. Classically, the terms “monoclonal, oligoclonal, polyclonal” are precisely used in the fields of immunology and cancer biology to address this issue, differentiating between the cases where a single clone, a few clones, or many clones, respectively, contribute to a process. In accordance with previous studies^{8,9}, we believe that the term “polyclonal” is more precise in this case and would help to address this concern. Indeed, Skulimowska et al.¹⁰ used a Confetti mouse to differentiate between two potential scenarios for bone marrow endothelial cell regeneration after radiation (oligoclonal regeneration vs. polyclonal regeneration) and found evidence for the latter. Similarly, we applied the term “polyclonal proliferation” or “polyclonal proliferation dynamics” throughout our manuscript, which is a more accurate description of the dynamics we have observed. In addition, we clearly defined our terminology in the introduction (lines 70-77) as requested by the reviewer and elaborated on these dynamics in lines 131-132, 153-155, 360-362, and the legend of **Fig. 1**.

References:

1. Clavreul, S. *et al.* Cortical astrocytes develop in a plastic manner at both clonal and cellular levels. *Nat Commun* **10**, 4884 (2019).
2. Rückert, T., Lareau, C. A., Mashreghi, M.-F., Ludwig, L. S. & Romagnani, C. Clonal expansion and epigenetic inheritance of long-lasting NK cell memory. *Nat Immunol* **23**, 1551–1563 (2022).
3. Ratz, M. *et al.* Clonal relations in the mouse brain revealed by single-cell and spatial transcriptomics. *Nat Neurosci* **25**, 285–294 (2022).
4. Liu, C. Y. *et al.* Wound-healing plasticity enables clonal expansion of founder progenitor cells in colitis. *Developmental Cell* **58**, 2309-2325.e7 (2023).

5. Kakiuchi, N. & Ogawa, S. Clonal expansion in non-cancer tissues. *Nat Rev Cancer* **21**, 239–256 (2021).
6. Maeda, H. & Kakiuchi, N. Clonal expansion in normal tissues. *Cancer Science* **115**, 2117–2124 (2024).
7. Wang, X. *et al.* Clonal expansion dictates the efficacy of mitochondrial lineage tracing in single cells. *Genome Biology* **26**, 70 (2025).
8. Cheung, K. J. *et al.* Polyclonal breast cancer metastases arise from collective dissemination of keratin 14-expressing tumor cell clusters. *Proceedings of the National Academy of Sciences* **113**, E854–E863 (2016).
9. Sadien, I. D. *et al.* Polyclonality overcomes fitness barriers in Apc-driven tumorigenesis. *Nature* **634**, 1196–1203 (2024).
10. Skulimowska, I. *et al.* Polyclonal regeneration of mouse bone marrow endothelial cells after irradiative conditioning. *Cell Rep* **43**, 114779 (2024).